# Microglial deficiency in the ATRX chromatin remodeler elicits a viral mimicry immune response that impacts neuronal function and behavior

Sarfraz Shafiq[1,2,3], Alireza Ghahramani[1,2,3○], Kasha Mansour[1,2,3○], Miguel Pena-Ortiz[2,3○], Julia K. Sunstrum[4,5], Milica Pavlovic[1,2,3], Yan Jiang[2,3], Megan E. Rowland[2,3], Wataru Inoue[4,5], Nathalie G. Bérubé[1,2,3,4]*

1 Department of Anatomy and Cell Biology, Schulich School of Medicine and Dentistry, Western University, London, Canada, 2 Department of Paediatrics, Schulich School of Medicine and Dentistry, Western University, London, Canada, 3 Division of Genetics & Development, Children's Health Research Institute, London, Canada, 4 Graduate Program in Neuroscience, Western University, London, Canada, 5 Robarts Research Institute, Western University, London, Canada

☀ These authors contributed equally to this work.
* nberube@uwo.ca

## Abstract

The importance of chromatin-mediated processes in neurodevelopmental and intellectual disability disorders is well recognized. However, how chromatin dysregulation in glial cells impacts cognitive abilities is less well understood. Here, we demonstrate that targeted loss of the ATRX chromatin remodeler targeted to microglia leads to altered cell morphology, increased chromatin accessibility profiles, and de-repression of endogenous retroelements, triggering viral mimicry. In mice that lack ATRX in microglia, CA1 hippocampal neuron morphology and electrophysiological properties are affected, and the mice display specific behavioral deficits. These findings demonstrate that ATRX is required in microglia to preserve chromatin structure and maintain microglial homeostasis. Disruption of these functions elicits neuroinflammation and may contribute to the pathology of human neurological disorders caused by *ATRX* mutations.

## Introduction

Microglia are the immune cells of the central nervous system and play critical roles in brain development, homeostasis, and disease. They modulate synaptic circuit remodeling, synaptic plasticity, and neuronal activity, which in turn can impact learning and memory [1–3]. Microglia are dynamically regulated and can exist in homeostatic or responding states depending on endogenous or exogenous stimuli [4]. Whereas resting microglia contribute to nervous system homeostasis and neuronal plasticity through the secretion of neurotrophic factors, microglial responsive states can induce

which permits unrestricted use, distribution, and reproduction in any medium, provided the original author and source are credited.

**Data availability statement:** All relevant data are included in the Supporting information and source data file. The RNA/ATAC sequencing data reported in this paper can be accessed at SRA accession number PRJNA787973. Data analysis and graphical representations were performed using R scripts and publicly available packages as denoted in the methods detail section. All relevant analysis scripts and pipelines used for RNA-seq, ATAC-seq, and ERV classification are available in a publicly accessible GitHub repository: https://github.com/aliireza96/Shafiq_PlosBiol_ATRX_microglia. The repository includes detailed documentation and instructions to reproduce all computational analyses. A snapshot of the repository has been archived on Zenodo with DOI: https://doi.org/10.5281/zenodo.15679297.

**Funding:** This work was supported by a Canadian Institutes of Health Research operating grant to N.G.B (FRN#183661). S.S. received a Children's Health Research Institute Trainee award funded by the Children's Health Foundation. K.M. received a Paediatrics Summer Studentship and a Paediatrics Graduate Studentship from the Department of Paediatrics at Western University. The above funders did not play a role in the study design, data collection and analysis, decision to publish, or preparation of the manuscript.

**Abbreviations:** aCSF, artificial cerebrospinal fluid; ATAC, assay for transposase-accessible chromatin; BH, Benjamini–Hochberg; cGAMP, cyclic guanosine monophosphate–adenosine monophosphate; cGAS, cyclic GMP-AMP synthase; DARs, differentially accessible regions; DCX, Doublecortin; DEGs, differentially expressed genes; DIV, days in vitro; dsRNA, double-stranded RNA; ERV, endogenous retrovirus; FANS, fluorescence-activated nuclei sorting; GSEA, Gene Set Enrichment Analysis; ID, intellectual disability; ISGs, interferon-stimulated genes; LTR, long terminal repeats; PRRs, pattern-recognition receptors; TF, transcription factor; TSS, transcriptional start sites; VST, variance stabilizing transformation.

neuronal dysfunction by producing neurotoxic factors and proinflammatory molecules [5,6]. Altered microglial states have been observed in neurodegenerative diseases such as Alzheimer's, Parkinson's, and Huntington's diseases and amyotrophic lateral sclerosis, as well as neurodevelopmental disorders including Down syndrome, autism spectrum disorder, and Rett syndrome [7–12]. This strongly suggests that active neuroinflammation may account for compromised neuronal survival, synaptic dysfunction, and cognitive deficits observed in these pathologies. Indeed, inhibiting reactive states of microglia has been shown to restore neuronal survival and cognitive deficits [10,13,14], highlighting the importance of microglia homeostasis in cognitive functions.

ATRX is a Snf2-type chromatin remodeler with crucial functions in the central nervous system [15–17]. In humans, *ATRX* mutations cause syndromic and non-syndromic intellectual disability (ID). In its syndromic form (ATR-X syndrome, OMIM #301040), patients also display seizures, microcephaly, hypomyelination, and severe developmental delays [18]. ID mutations are largely located in two hotspots: the DNMT3A/B and DNMT3L (ADD) globular domain and the Snf2-like enzymatic domain [19,20]. One of the major protein interactors of ATRX is DAXX, which is a histone chaperone for the variant histone H3.3. Together, ATRX and DAXX incorporate H3.3 at repetitive regions of the genome, including telomeres, pericentric repeats, rDNA repeats, and endogenous retroviral elements [21–24].

We previously reported that targeted deletion of *Atrx* in postnatal neurons of the mouse forebrain causes long-term spatial and associative memory deficits, in parallel with hippocampal structural alterations and impaired hippocampal synaptic transmission [17,25]. In the present study, we show that *Atrx* deletion in microglia of the mouse central nervous system impacts chromatin accessibility and gene expression profiles in these cells. Our results show that in microglia, ATRX limits DNA damage and the expression of retroelements. Upon loss of ATRX, microglia morphology changes and the DNA and RNA sensing pathways are activated, leading to an interferon response and cytokine release. These changes are associated with morphological and electrophysiological changes in hippocampal CA1 neurons, memory deficits, and reduced anxiety. These findings reveal the deleterious effects of ATRX deficiency in microglia on chromatin structure and the resulting non-cell autonomous harmful effects on neurons and cognitive processes.

## Results

### Altered morphology and increased CD68 foci in ATRX-null microglia

We targeted *Atrx* deletion in microglia by mating tamoxifen-inducible *Cx3cr1*[ERT2] mice [26] and *Atrx*[LoxP] mice [16], resulting in male progeny lacking ATRX in microglia (hereon referred to as ATRX miKO mice). The nuclear membrane Sun1GFP reporter gene (IMSR_JAX:021039) was also introduced to track the fate of Cre-expressing microglia [27]. Cre expression was induced by daily intraperitoneal injections of tamoxifen for five consecutive days starting at postnatal day (P)45 (Fig 1A). There was no difference in body weight between control and ATRX miKO mice at 2 and 3 months of age, nor was there a difference in *Cre* and *Sun1GFP* expression by

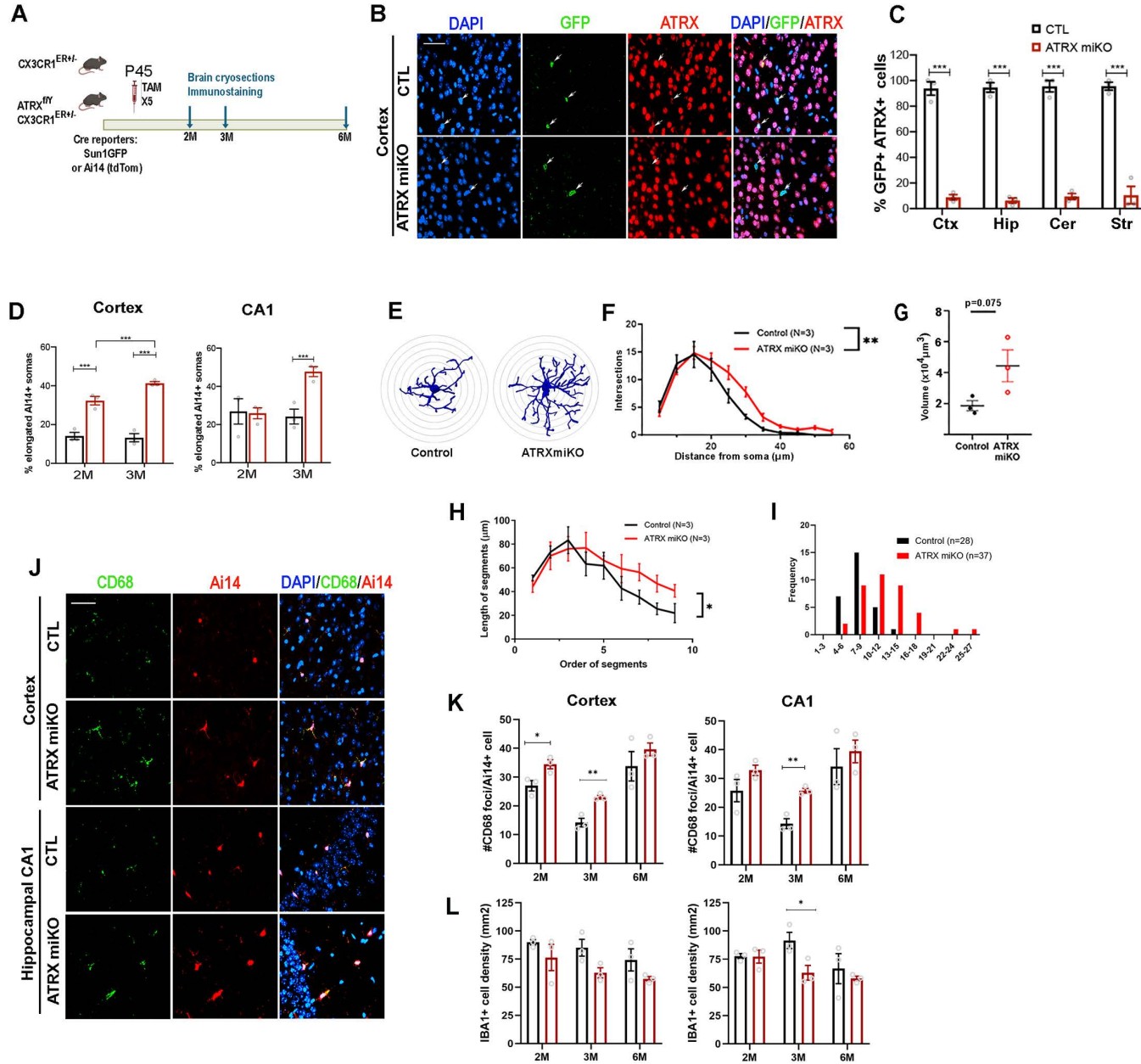

**Fig 1. Altered morphology and increased CD68 foci in ATRX-null microglia. (A)** Schematic representation of the experimental approach. **(B)** Immunofluorescence images of anti-GFP (green) and anti-ATRX (red) counterstained with DAPI (blue) illustrating absence of ATRX in Sun1GFP+ microglia nuclei in 2-month-old ATRX miKO mice. Scale bar, 50 μm. **(C)** Quantification of ATRX knockout efficiency in different brain regions ($n=3$ each genotype, cortex $p=0.0001$, hippocampus $p=0.00003$, cerebellum $p=0.0001$, striatum $p=0.0003$). **(D)** Quantification of microglia with elongated soma in cortex and hippocampal CA1 region (2 months: cortex $p=0.003$, CA1 $p=0.007$; 3 months: cortex $p=0.0002$, CA1 $p=0.007$). **(E)** Representative images of microglia tracings. **(F)** Sholl analysis reveals excessive branching in ATRX miKO compared to control microglia. Two-way ANOVA Fgenotype $(1,44) = 8.857$, $p=0.0047$. **(G)** Convex hull volume analysis of control and ATRX-null microglia. **(H)** Increased segment length in ATRX-null microglia. Cut-off branching order of 9. Two-way ANOVA $F(1,36)=4.570$, $p=0.0394$. **(I)** Average number of microglia for each maximum branching order arranged in bins. For panels E–I, $n=8$–15 microglia per mouse, $N=3$ mice per genotype were analyzed. **(J)** Immunofluorescence staining of CD68 (green) and Ai14+ (red) cortical and hippocampal CA1 microglia. Scale bar, 50 μm. **(K)** Number of CD68 foci per Ai14+ microglia in cortex and hippocampal CA1 ($n=3$ each genotype, 2 months cortex $p=0.036$, 3 months cortex $p=0.005$; 6 months cortex $p=0.349$; 2 months CA1 $p=0.174$, 3 months CA1 $p=0.004$, 6 months CA1 $p=0.512$. **(L)** IBA1+ microglia density in cortex and CA1 ($n=3$ each genotype, cortex 2 months $p=0.320$, 3 months $p=0.062$, 6 months $p=0.170$; CA1 2 months $p=0.941$, 3 months $p=0.041$, 6 months $p=0.550$). Student $T$ test. Ctx, cortex; Hip, hippocampus; Cer, cerebellum; Str, striatum. The data underlying this figure can be found in the S1 Data file.

quantitative reverse transcription polymerase chain reaction (RT-PCR), suggesting equivalent level of recombination in mice of both genotypes (S1A and S1B Fig). The efficiency of Cre recombination in microglia was evaluated by quantifying the number of cells expressing Sun1GFP and the microglial marker Ionized calcium binding adaptor molecule 1 (Iba1), revealing over 95% overlap in control and ATRX miKO mice across multiple brain regions (S1C Fig). Assessment of ATRX protein expression by immunofluorescence staining of brain sections shows that more than 90% of Sun1GFP+ cells lack ATRX expression across various brain regions in ATRX miKO mice (Fig 1B and 1C).

We next replaced the Sun1GFP with the tdTomato-Ai14 Cre-sensitive allele to allow morphological visualization of control and ATRX-null microglia in brain sections. Upon microscopic examination of tdTomato fluorescence [28], we initially noted that the soma of microglia lacking ATRX appear morphologically altered compared to control microglia. We categorized microglia soma structure as round, intermediate, or elongated (S1D Fig), and quantified each category and measured aspect ratios. The data confirms that ATRX-null microglia somas in the cortex are more elongated at 2 months, with a larger effect seen at 3 months (Figs 1D, S1E, and S1F). The nuclei themselves are also larger and appear rod-shaped in ATRX-null compared to control microglia in the cortex (S1G and S1H Fig). Hippocampal microglia somas and nuclei (CA1, CA2, CA3, and DG regions) are rod-shaped, but this only became evident at 3 months of age (Figs 1D and S1I). We next performed a more detailed analysis of microglia morphology using tracings of confocal z-stack images (Fig 1E). We detected a significant increase in branching of microglia processes in ATRX miKO mice, increased microglia convex hull volume, and increased length of outer projections (Fig 1F–1H). The maximum order of branching reached in the ATRX null microglia was also higher than in controls (Fig 1I). Overall, these results show that ATRX-null microglia are larger and hyper-ramified, similar to microglia exposed to the cytokine Interferon α [29,30].

We considered that the change in morphology of ATRX-null microglia might reflect a reactive state sometimes associated with upregulation of the lysosomal Cluster of Differentiation 68 (CD68) [4,31]. Analysis of stained brain cryosections shows that the number of CD68 foci in microglia is significantly increased in the cortex of ATRX miKO mice at 2 and 3 months of age, and that this effect has largely resolved by 6 months of age (Fig 1J and 1K). In hippocampal regions, CD68 staining was only significantly increased at 3 months in ATRX miKO mice (Figs 1K and S1J). We note that CD68 control levels are higher at 2 months compared to 3 months, although the expectation is of a progressive increase as the mice age. One possible explanation is a transient effect of injections performed at P45–P49. The density of microglia is largely maintained in the cortex and hippocampus of ATRX miKO mice, except for a decrease at 3 months of age (Figs 1L and S1K). Overall, these results indicate that loss of ATRX expression in microglia alters microglia morphology and leads to a transient increase in CD68, indicating a loss of homeostasis.

## Transcriptome analysis reveals changes in cell proliferation, genome integrity, and activation of the innate immune response

We next evaluated whether transcriptomic changes in ATRX-null microglia might provide insight into specific abnormalities underlying the observed morphological changes. Sun1GFP+ microglia nuclei were obtained from the cortex and hippocampus of control and ATRX miKO mice by fluorescence-activated nuclei sorting (FANS; Fig 2A) [27,32]. Given that both control and ATRX miKO undergo FANS simultaneously, we assume that any effects on gene expression caused by the method itself would occur in cells of both genotypes and would not appear as differentially expressed. The method also bypasses several problematic artifacts: it doesn't include cell dissociation steps that rupture cell processes, or nuclei staining that could affect chromatin structure. RNA-seq of sorted microglia identified 6,168 differentially expressed genes (DEGs) between control and ATRX-null microglia, 3,265 of which were upregulated, and 2,903 downregulated (Adj $p < 0.05$, S1 Table). Gene Set Enrichment Analysis (GSEA) of the upregulated DEGs identified biological process categories mainly related to cell proliferation (sister chromatid segregation, DNA replication, nuclear division, mitotic cell cycle), DNA damage repair (DNA metabolic processes, DNA repair, cellular response to DNA damage stimulus), and to the immune response (defence response) (Fig 2B and S2 Table). For downregulated DEGs, the top 10 enriched biological

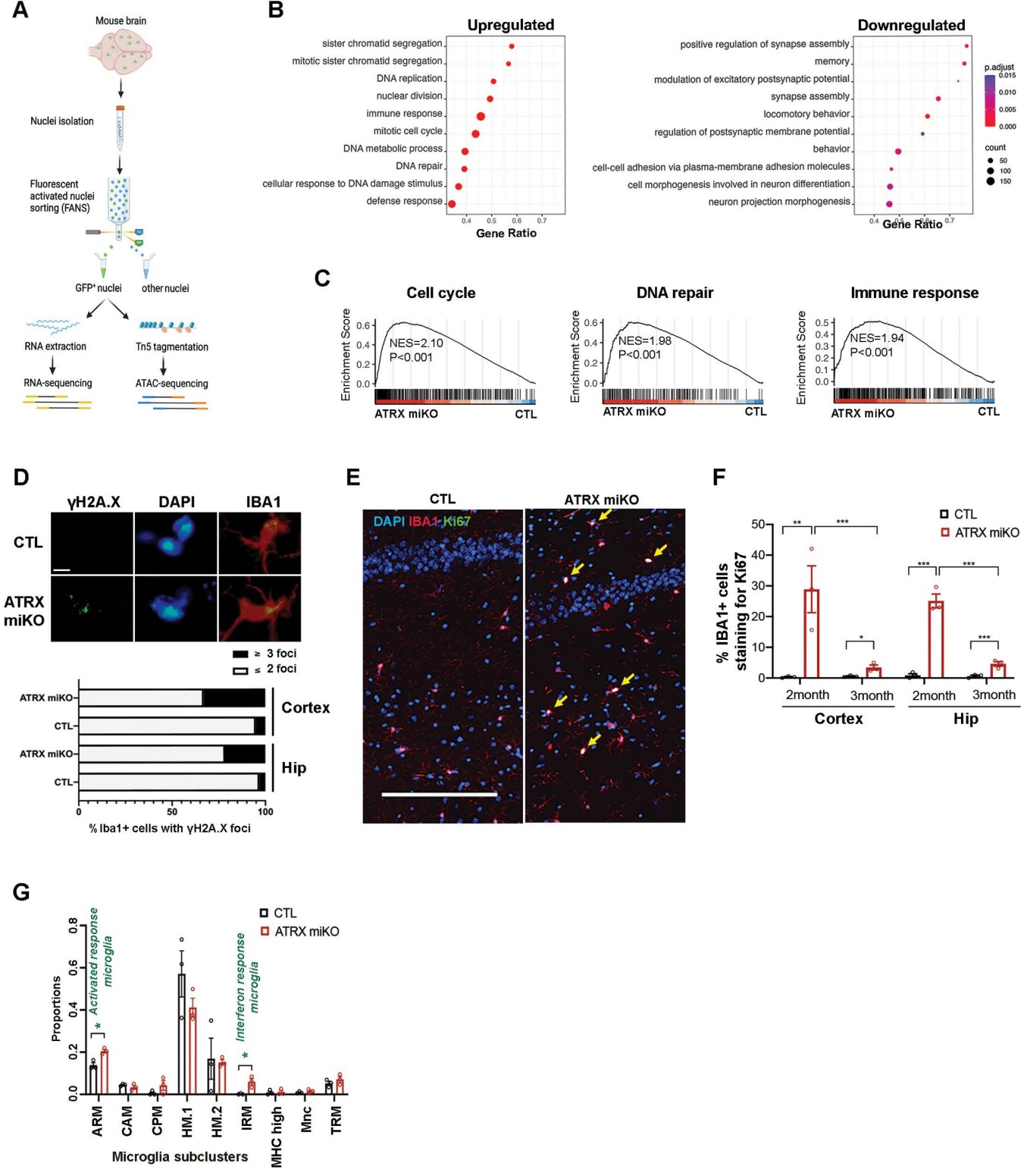

**Fig 2. Ablation of ATRX in microglia leads to increased proliferation, DNA damage, and immune activation. (A)** Schematic illustration of fluorescence-activated microglia nuclei sorting from the mouse cortex followed by RNA-sequencing. **(B)** The top 10 functionally enriched pathways for upregulated and downregulated genes. **(C)** Gene Set Enrichment Analysis reveals enrichment of genes linked to cell cycle, DNA repair, and immune

activation in ATRX-deficient microglia. NES, normalized enrichment score. **(D)** Immunofluorescence staining of IBA1 and γ-H2AX in the cortex of control and ATRX miKO mice. Scale bar, 10 μm. Graph below depicts γ-H2AX foci per microglia in the cortex and hippocampus ($n = 3$ each genotype, cortex $p = 0.003$, hippocampus $p = 0.018$, Student $T$ test). **(E)** Immunofluorescence staining of IBA1 and Ki67 in the hippocampus of control and ATRXmiKO mice at 2 months of age. Scale bar, 200 μm. **(F)** Quantification of Ki67-positive microglia in the cortex and hippocampus reveals increased proliferation of ATRX-null microglia ($n = 3$ each genotype, Ki67 2-month cortex $p = 0.020$, 3-month cortex $p = 0.025$; 2-month hippocampus $p = 0.0005$, 3-month hippocampus $p = 0.009$, Student $T$ test). **(G)** Deconvolution of bulk microglia nuclei RNA-seq into single-cell identify sub-clusters. ARM, activated response microglia; CAM, CNS-associated macrophages; CPM, cycling and proliferating microglia; HM.1, homeostatic microglia cluster 1; HM.2, homeostatic microglia cluster 2; IRM, interferon-response microglia; MHC.high, high MHC-expressing microglia; Mnc, monocytes; TRM, transitioning microglia. *$p < 0.05$, Student $T$ test. Error bars represent SEM. The single cell data used for deconvolution was downloaded from the GEO database (GSE142267; Sierksma and colleagues, EMBO Mol Med 2020, 12(3), e10606). The data underlying this figure can be found in the S1 Data file.

processes are related to synaptic transmission, neuronal morphogenesis, cell adhesion, and memory (Fig 2B). GSEA enrichment plots demonstrate that upregulated genes involved in the cell cycle (i.e., *Ccnb1, Ccnd3, E2F2, E2F3, E2F7, E2F8, Cdc25c, Mcm10*), DNA repair (mostly homologous recombination repair) and replication stability (i.e., *Rad51, Brc1a, Fanca, Gen1, DNA2, Blm, DNA2*) and the immune response and inflammatory signaling (i.e., *Cd69, Cd72, Axl, Mx1, Mx2, Tnf, Stat2, Cxcl10*), were significantly over-represented in ATRX-null compared to control microglia (Fig 2C). To validate these results, we performed immunofluorescence staining for γH2AX, a marker of DNA double-stranded breaks and Ki67, a marker of cell proliferation. The results confirm that ATRX-null microglia contain more DNA damage (Fig 2D). There was also a notable increase in proliferative Ki67+ ATRX-null microglia in the cortex and hippocampus at 2 months that largely (but not completely) resolves by 3 months of age (Fig 2E and 2F). Deconvolution analysis of the RNA-seq data was performed to estimate the relative proportions of microglia subpopulations in control and ATRX-null microglia based on published single-cell data [33]. Of the nine microglia subclusters identified, the inferred proportion of "activated response" and "interferon response" microglia subclusters is increased in ATRX miKO mice compared to controls, seemingly at the expense of homeostatic microglia, again suggesting that ATRX deletion causes an immune response in microglia (Fig 2G). A limitation of this in silico approach, however, is the inability to detect microglial cell states not represented in this published dataset.

## Increased chromatin accessibility in ATRX-null microglia

To investigate whether the transcriptional changes upon loss of microglial Atrx are associated with altered chromatin accessibility, Sun1GFP⁺ microglia nuclei from the cortex and hippocampus were sorted and subjected to the assay for transposase-accessible chromatin followed by sequencing (ATAC-seq) (Fig 3A) [32]. Two different approaches, MACS2-DESeq2 and csaw-EdgeR identified a similar number of DARs with comparable genomic distribution (S2A and S2B Fig). We proceeded with the list of DARs identified using MACS2 for downstream analysis. A total of 32,936 DARs were identified (Adj $p < 0.05$), located within gene promoters, exons, introns, downstream regions, and distal intergenic regions (S2A Fig and S3 Table), the majority (94%) displaying increased accessibility in ATRX-deficient microglia compared to controls (Fig 3A). Accessibility heatmaps and violin plots of DARs display a marked increase in chromatin accessibility at genes, especially at the transcriptional start sites (TSS) (Fig 3B and 3C).

Intersection of the ATAC-seq and transcriptomic data shows that 2,945 of the total 6,182 DEGs (~47%) have a DAR at their TSS (Fig 3D), demonstrating that altered chromatin accessibility is often coupled with transcriptional effects. We also find that increased chromatin accessibility is associated with elevated expression of genes related to the immune response (Fig 3E), cell cycle, and DNA repair (S2C Fig). Within the immune response category, DEGs mainly belong to the innate immune response, interferon and cytokine signaling pathways, and a large majority of DEGs are significantly upregulated and their chromatin more accessible in ATRX-null microglia (Fig 3E).

We next characterized the transcription factor (TF) binding site profiles represented in the ATAC-seq data. We scanned all the annotated TF motifs and scored their differential binding using TOBIAS [34]. We identified 30 TFs with increased

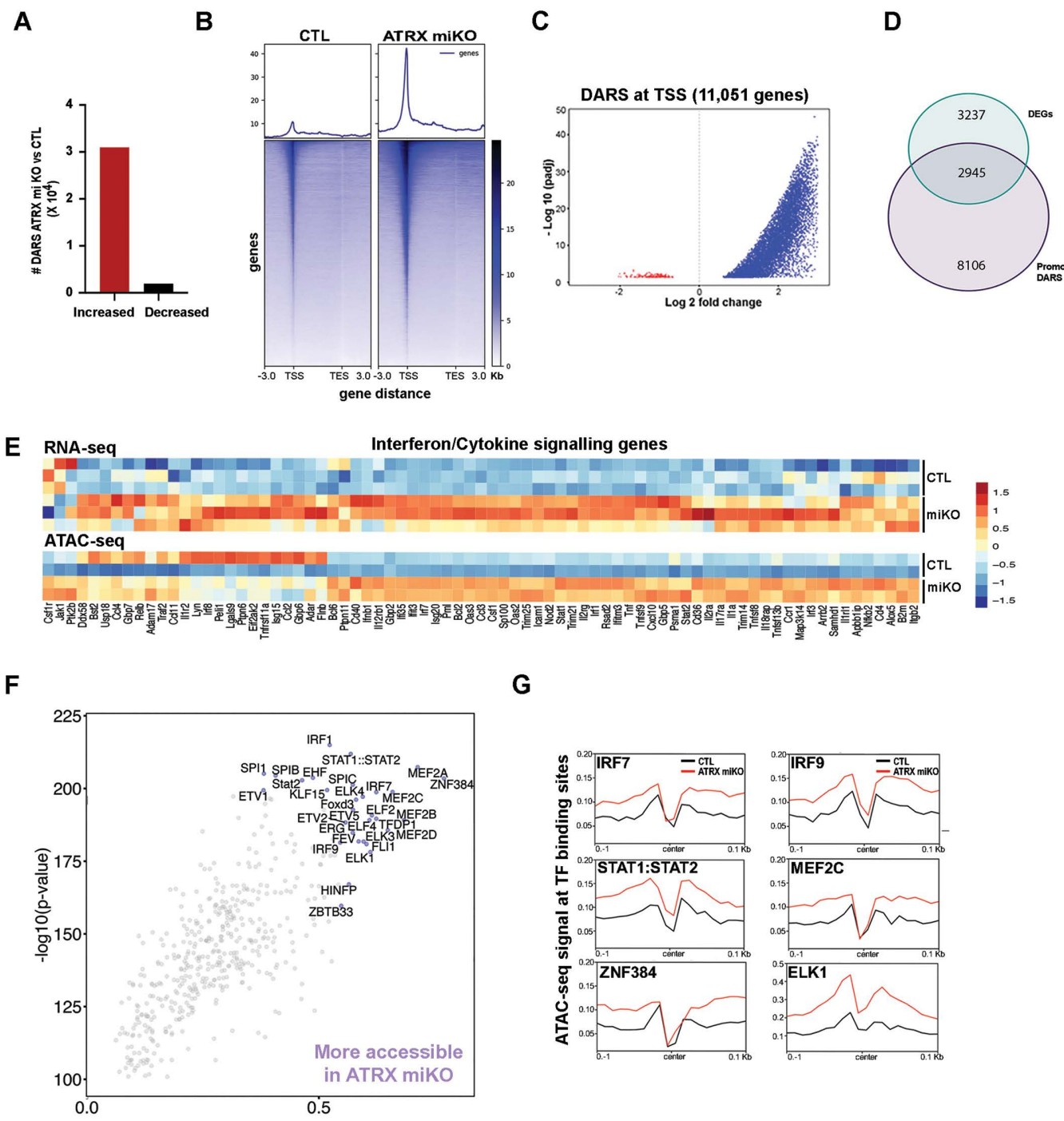

**Fig 3. Chromatin accessibility is increased at interferon response/cytokine genes and at interferon signaling TF binding motifs. (A)** More genomic regions exhibit increased (red) vs decreased (black) chromatin accessibility in ATRX-null compared to control sorted microglia. **(B)** Heatmap of chromatin accessibility differences between control and ATRX-null microglia highlights general increased accessibility at promoters and gene bodies in ATRX-null microglia. **(C)** Violin plot of all genes where DARS are localized at the TSS. **(D)** Venn diagram showing the extent of overlap between DEGs identified by RNA-seq and DARs at gene promoters. **(E)** Heatmap of RNA-seq and ATAC-seq data highlights correlation between gene expression and chromatin accessibility changes at interferon and cytokine signaling genes. **(F)** TF binding motif enrichment analysis from increased DARS. Top 2% are labeled. **(G)** Genome-wide enrichment of interferon-related TF binding motifs in DARs identified by ATAC-seq. Stronger dips in the center indicate higher confidence in binding, and the height of the nearest summit to the center indicates chromatin accessibility. DAR, differentially accessible region; DEG, differentially expressed gene; TSS, transcription start site; TES, transcription end site; CTL, control. The data underlying this figure can be found in the S1 Data file.

differential binding scores (top 2%) between control and ATRX miKO (Fig 3F and S4 Table). The accessibility of binding sites for TFs related to interferon signaling (IRF1, IRF7, IRF9, STAT1, STAT2, ELK1, MEF2C, and ZNF384) was increased ATRX miKO (Fig 3G), confirming broad instigation of an immune and interferon response in ATRX-null microglia.

## Deletion of ATRX in microglia activates endogenous retroelements

Failure to suppress retroelements in the genome is known to trigger an immune response [35]. Given ATRX's role in heterochromatin formation and previous reports of ATRX binding to retrotransposons in mouse embryonic stem cells [22,36], we investigated whether activation of the immune response in ATRX-null microglia is linked to the de-repression of retroelements. The analysis of the ATAC-seq data identified 11,155 DARs (Adj $p < 0.05$) in repetitive sequences across the genome between control and ATRX-null microglia, with >99% exhibiting increased accessibility (Fig 4A and S5 Table). These mainly belong to GC/C-rich repeats, simple repeats as well as long terminal repeats (LTR), especially ERVK and ERVL as well as non-LTR transposable elements (LINES and SINEs) (Fig 4B). From the RNA-seq data, we identified 20,189 differentially expressed retroelements (Adj $p < 0.05$) between control and ATRX miKO (S6 Table). Consistent with ATAC-seq, the majority of these were de-repressed in ATRX-null microglia (71%) and mainly belong to LTR class ERVK and ERVL, as well as LINEs and SINEs (Fig 4C and 4D). Applying RetroTector to our RNA-seq data revealed that a notable subset of the upregulated ERVs in ATRX-null microglia are full-length ERVs (3%, $n = 99$) and the majority (83%, $n = 2,641$) are solo LTRs (S6 Table). TE subfamily-based analysis was also performed and confirmed a strong de-repression of the LTR class (Figs 4E and S3A and S7 Table). Conversely, the deletion of ATRX in oligodendrocytes or astrocytes does not cause such an extensive de-repression of retroelements suggesting that retroelement suppression is particularly sensitive to ATRX loss in microglia (S3B and S3C Fig and S8 Table). Overall, sequencing analysis demonstrates that loss of ATRX in microglia leads to a marked de-repression of TEs.

## Loss of ATRX engages a viral mimicry response

The expression of genes related to a viral response is significantly enriched in ATRX-null microglia and correlates with increased chromatin accessibility (Fig 4F and 4G). Both de-repression of retroelements and DNA damage have been shown to trigger a viral mimicry response through sensing of cytosolic dsRNA or dsDNA, respectively [37–40]. Bidirectional transcription of retroelements could result in the formation of double-stranded RNA (dsRNA) [41,42] that trigger the interferon response [43,44]. We identified 4,480 and 5,226 significant differentially expressed TEs (P-adjusted value < 0.05) for forward and reverse analysis, respectively, with 57 TEs significantly expressed from both strands (S9 Table). Examples of LINEs, LTRs, and SINES with increased sense and antisense transcripts in ATRX miKO microglia are shown in Figs 4H and S2D.

To determine whether excessive expression of retrotransposons results in dsRNA species that reach the cytoplasm, we established primary cell cultures and performed immunofluorescence staining with a J2 antibody that specifically detects dsRNA [45]. This revealed an accumulation of dsRNA in Ai14+ATRX-null microglia (Fig 4I). To identify some of the TE-derived dsRNAs that reside in the cytoplasm, RNA immunoprecipitation using J2-conjugated beads was performed. RT-qPCR analysis of top upregulated TEs (ERVB2_1-I MMdup5 and MLT1C_LTR-ERVL) showed that they are enriched in the cytoplasmic fractions of ATRX miKO but not control brain tissue (Fig 4J). ERVB2 is evolutionary younger and has a higher potential to directly induce a strong viral mimicry response due to possible transcription of longer, complementary RNA sequences.

Increased DNA damage and cytoplasmic dsRNA can engage the viral mimicry pathway. Indeed, multiple genes belonging to the DNA/RNA sensing pathways are overexpressed in ATRX-null microglia, including Cyclic GMP-AMP synthase (cGAS) and Retinoic acid-inducible gene I (RIG-1/*Dhx58*) (S4A Fig). cGAS senses dsDNA and produces cyclic guanosine monophosphate–adenosine monophosphate (cGAMP), which then binds Stimulator of interferon response CGAMP interactor 1 (STING1) [46]. RIG-1 senses dsRNA and interacts with the Mitochondrial antiviral-signaling protein (MAVS) [46].

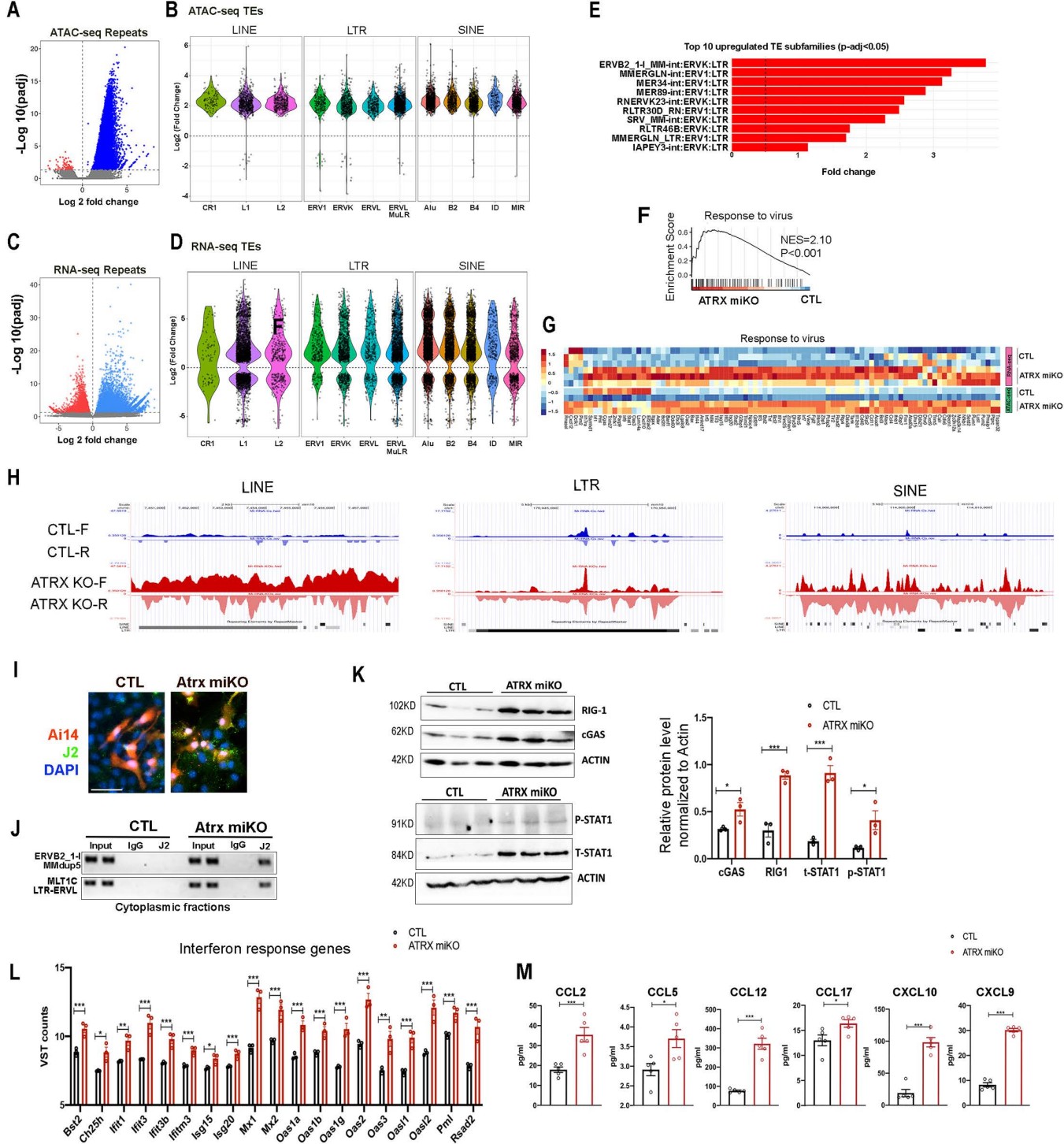

**Fig 4. De-repression of endogenous retroelements triggers viral mimicry, interferon, and cytokine response. (A)** Volcano plot of ATAC-seq results for sequence repeats with increased (blue) and decreased (red) chromatin accessibility in ATRX-null microglia. Data points in gray did not meet the significance threshold. **(B)** Differential chromatin accessibility between control and ATRX-null microglia for different classes of retroelements. 11,133 elements are more accessible and 22 are less accessible. Only TEs with padj > 0.05 are displayed. **(C)** Volcano plot showing retroelement loci with increased (blue) and decreased (red) transcript levels in ATRX-null compared to control microglia. Data points in gray did not meet the significance threshold. **(D)** Differential expression for each class of retroelements between ATRX miKO and control sorted microglia nuclei. 14,250 elements are

upregulated and 5,939 are downregulated. Only TEs with padj > 0.05 are displayed. **(E)** List of most upregulated TE subfamilies. **(F)** GSEA reveals that the "response to virus" pathway is significantly enriched in ATRX-null microglia. NES, normalized enrichment score. **(G)** Heatmap representing transcript levels and chromatin accessibility for the "response to virus" pathway. **(H)** UCSC genome browser tracks showing bi-directional transcription of retroelements belonging to LINE, LTR, and SINE in sorted ATRX miKO microglia nuclei. F and R represent forward and reverse strands, respectively. **(I)** Staining of primary mixed glial cultures obtained from control and ATRX miKO cortex with the anti-dsRNA J2 antibody at 7 days in vitro. Scale bar, 50 μm. **(J)** RNA immunoprecipitation of cytoplasmic cerebellar fractions with the J2 antibody followed by RT-PCR of two upregulated retroelements. **(K)** Western blots of DNA/RNA sensing pathway components and graph of relative protein levels ($n = 3$ each genotype; cGAS $p = 0.047$, RIG-1 $p = 0.002$, pSTAT1 $p = 0.042$, STAT1 $p = 0.001$, Student $T$ test). Error bars represent ±SEM. **(L)** Gene expression of interferon-stimulated genes (ISGs). Variance stabilizing transformation (VST) counts from RNA-seq data ($n = 3$ each genotype; Bst2 $p = 0.007$, Ch25h $p = 0.022$, Ifit1 $p = 0.012$, Ifit3 $p = 0.003$, Ifit3b $p = 0.006$, Ifitm3 $p = 0.004$, Isg15 $p = 0.037$, Isg20 $p = 0.009$, Mx1 $p = 0.006$, Mx2 $p = 0.006$, Oas1a $p = 0.001$, Oas1b $p = 0.007$, Oas1g $p = 0.002$, Oas2 $p = 0.002$, Oas3 $p = 0.015$, Oasl1 $p = 0.003$, Oasl2 $p = 0.004$, Pml $p = 0.007$, Rsad2 $p = 0.006$, Student $T$ test). Error bars represent ±SEM. **(M)** Cytokine/chemokine levels ($n = 5$ each genotype; CCL2 $p = 0.001$, CCL5 $p = 0.020$, CCL12 $p = 0.000$, CCL17 $p = 0.032$, CXCL10 $p = 0.000$, CXCL9 $p = 0.000$, Student $T$ test). The data underlying this figure can be found in the S1 Data file.

Western blot analysis shows increased RIG-1 in the ATRX miKO cortex, hippocampus, and cerebellum in 2- and 3-month-old mice; cGAS is increased in the cortex and hippocampus at 2 months but not at 3 months while it is increased at both time points in the cerebellum of ATRX miKO mice (Figs 4K, S4B, and S4C). These results suggest a potential activation of the DNA- and RNA-sensing pathways in ATRXmiKO brains.

The cGAS or RIG-1 signaling cascades both lead to the phosphorylation and heterodimerization of the Signal transducer and activator of transcription 1 and 2 (STAT1, STAT2) [47]. *Stat1* gene expression is significantly increased in ATRX-null microglia (S4A Fig), and we show that total STAT1 (t-STAT1) and phospho-STAT1 (p-STAT1) protein levels are increased in ATRX miKO cortical tissue compared to controls (Fig 4K). Downstream of STAT activation, the expression of numerous interferon-stimulated genes (ISGs) was increased in ATRX-null microglia (Fig 4L).

Recognition of nucleic acids by pattern-recognition receptors (PRRs), such as RIG-1 and cGAS, is essential for triggering an antiviral immune response by inducing the production of proinflammatory cytokines [48]. Given that several cytokine genes were upregulated in ATRX-null microglia nuclei, a cytokine/chemokine array analysis of cortical protein extracts was performed and corroborates elevated levels of numerous cytokines/chemokines in ATRX miKO mice, including CC motif ligand 2 (CCL2), CC motif ligand 5 (CCL5), CC motif ligand 12 (CCL12), CC motif ligand 17 (CCL17), C-X-C motif chemokine ligand 9 (CXCL9), and C-X-C motif chemokine ligand 10 (CXCL10) (Fig 4M and S10 Table). Note; however, that the extent of cytokine production detected by this assay could be amplified due to secondary astrocytosis. These data show that loss of ATRX in microglia activates the DNA and RNA-sensing pathways as well as an interferon and cytokine response.

### Loss of microglial ATRX impacts the morphology and electrophysiological states of hippocampal CA1 neurons

Given that microglia sense neural activity and, in turn, can modulate neuronal morphology and activity [49], we next evaluated the impact of ATRX deficiency in microglia on hippocampal CA1 neurons. We introduced the Thy1GFP-M allele [50] in our model to achieve sparse fluorescence labeling of neurons, facilitating morphological assessments. Sholl analysis of CA1 neuronal tracings reveals significantly increased dendritic branching (Fig 5A) and increased density of filopodia spines in mid-apical dendrites (Fig 5B), with no difference in overall dendrite length (Fig 5C) when comparing control and ATRX miKO CA1 neurons. Since loss of homeostasis in microglia can influence dentate gyrus stem cells, we also looked for potential deleterious effects on this cell population at 3 months of age. No difference was detected in dentate granule progenitor proliferation, differentiation, or apoptosis, as measured by Ki67, Doublecortin (DCX), and activated caspase 3 staining (S5A–S5C Fig).

We then evaluated the neuronal membrane properties (i.e., intrinsic properties) as well as excitatory synaptic transmission in hippocampal CA1 pyramidal neurons. We first studied the subthreshold properties by artificially injecting a small-amplitude current in whole-cell current clamp recordings (Fig 5D). We found that CA1 neurons from ATRX miKO

mice exhibit a lower membrane resistance (input resistance) compared to control mice (Fig 5E). A decrease of the input resistance can be due to an increase in the cell size (surface membrane area [51]). However, the membrane capacitance (Cm), a proxy for the surface membrane area, was similar between the ATRX miKO and control (Fig 5F), suggesting that CA1 neurons lacking ATRX have increased opening in membrane channels with little, if any, change in cell size. In response to the steps of depolarizing current injections, we observed a regular-spiking pattern as reported elsewhere for hippocampal pyramidal neurons [52] in both control and ATRX miKO mice (Fig 5G). The overall current-spike frequency relationship is similar between control and ATRX miKO mice (Fig 5H). Other firing properties, including latency and inter-spike interval, are also not different between control and ATRX miKO mice (Fig 5I). The examination of the action potential shapes elicited by the minimal current injection (rheobase) revealed that CA1 neurons from ATRX miKO mice exhibit an increased action potential amplitude compared to control mice (Fig 5J and 5K). The rheobase current and threshold were not different between control and ATRX miKO neurons (Fig 5L and 5M). We also studied spontaneous excitatory postsyn-aptic current (sEPSC) in voltage-clamp recordings (Fig 5N). The frequency was not different between control and ATRX miKO mice (Fig 5O). However, ATRX miKO mice showed a significant leftward shift in the sEPSC amplitude distribution (Fig 5P). The average sEPSC amplitude change did not reach a statistically significant decrease (Fig 5Q), suggesting that the synaptic change may occur in subpopulations of synapses and their amplitude changes may be masked when expressed as population average. These results demonstrate that microglial ATRX deficiency imparts noncell autonomous effects on CA1 neurons, altering their morphology and electrophysiological properties.

## Loss of ATRX in microglia leads to memory deficits and anxiolytic effects in mice

We next assessed whether the molecular and cellular defects observed in ATRX-null microglia impact cognitive abilities. Details of statistical analyses of all behavior assays can be found in S12 Table. In the open field test, ATRX miKO and control mice travel an equivalent distance (Fig 6A), indicating that general locomotor activity is not affected. The total time spent in the center also does not differ between genotypes (Fig 6B).

Several anxiety tests were conducted, revealing noteworthy differences between the ATRX miKO and control groups. In the light-dark paradigm, ATRX miKO mice spend significantly more time in the lit area compared to control mice, suggest-ing reduced anxiety levels (Fig 6C). This finding is further supported in the elevated plus maze test during which ATRX miKO mice spend significantly more time in the open arm than control mice (Fig 6D). However, there is no significant difference between groups in the number of entries into different arms of the elevated plus maze (S6A Fig).

In the contextual fear conditioning test, ATRX miKO mice spend the same amount of time freezing as control mice (Fig 6E and 6F) and in the Y-maze, the percent spontaneous alternations and total arm entries were not different between genotypes (Fig 6G), indicating that contextual fear memory and working memory processes are intact in ATRX miKO mice.

In the Morris water maze task, ATRX miKO and control mice had equivalent latency to find the platform as well as speed and distance traveled over the four days of training (Figs 6H, S6B, and S6C). Short-term and long-term spatial memory were tested on day 5 and day 12, respectively.

On day 5, control and ATRX miKO mice spent significantly more time in the target compared to other quadrants, sug-gesting intact short-term spatial memory. However, the 12-day probe trial revealed that ATRX miKO mice have impaired long-term spatial memory. A two-way ANOVA revealed a significant interaction of genotype and time spent in the target versus non-target quadrants, and a post hoc test identified a significant difference in percent time spent in the target quad-rant between control and ATRX aiKO mice (Fig 6I).

Finally, mice were evaluated in the novel object recognition memory test. During the habituation period, control and ATRX miKO mice equally explored the two identical objects (Fig 6J). When memory was tested (1.5 and 24 h), the effect of the ATRX miKO genotype on memory was only significant after 24 h, indicating a long-term memory deficit. However, when the data is expressed as discrimination index, the difference between ATRX miKO and control mice is not significant

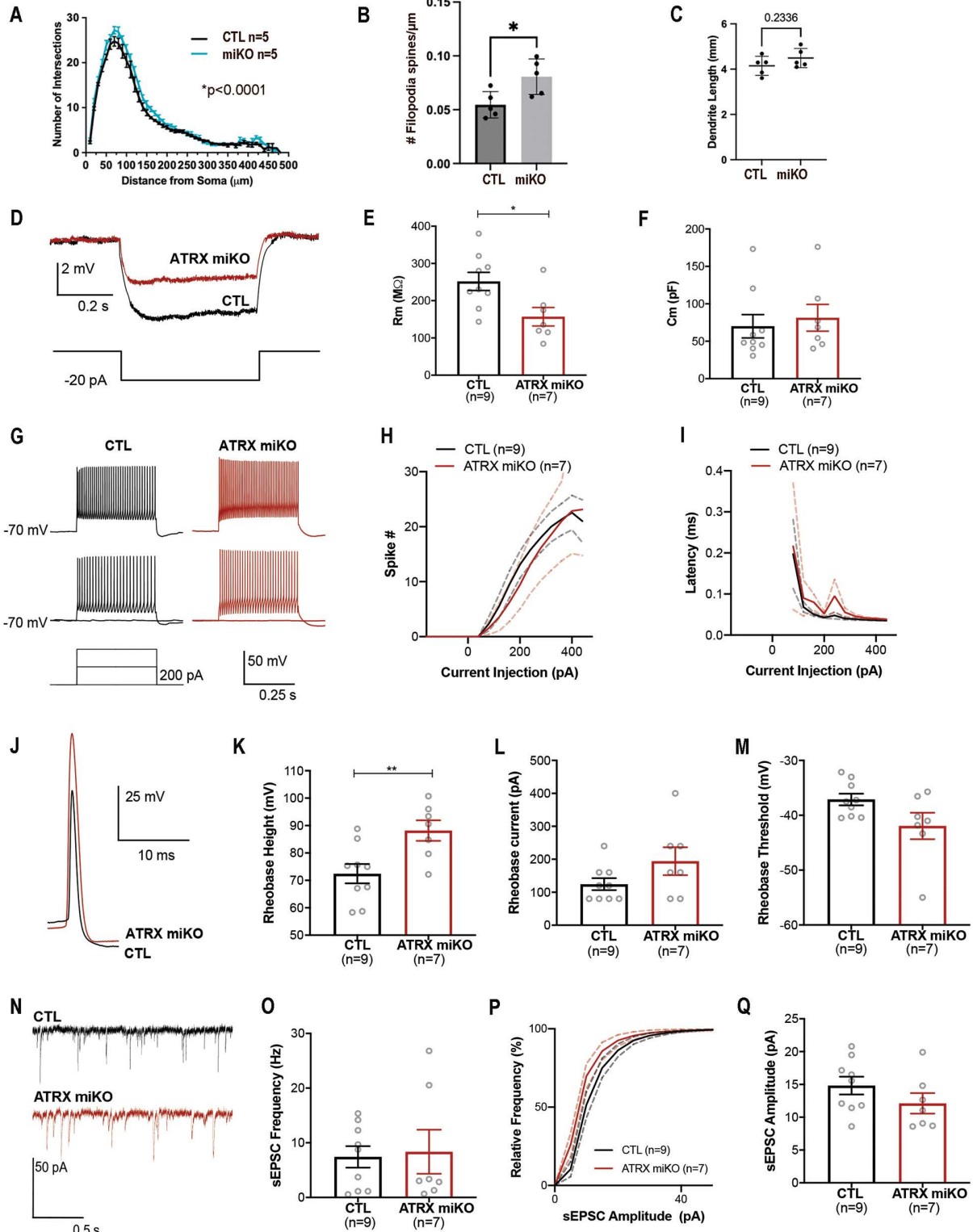

**Fig 5. Altered morphology and electrophysiological properties of CA1 hippocampal pyramidal neurons in ATRX miKO mice. (A)** Sholl analysis of hippocampal CA1 neurons, $n = 8$ neurons per mice, $N = 5$ mice each genotype. Two-way ANOVA was applied. **(B)** Density of filopodia spines per segment length (µm). For each mouse, five separate dendritic segments were measured for six individual neurons. $N = 5$ mice per genotype. Student *T* test,

$p=0.0220$. **(C)** CA1 neuron dendrite length in control and Atrx miKO mice. Student $T$ test, $p=0.2336$. **(D)** Representative current clamp traces of −20 pA step. The step protocol is shown at the bottom. **(E, F)** Summary of membrane resistance (Student $T$ test, $p=0.017$) and capacitance (Student $T$ test, $p=0.3266$) calculated from the −20 pA step in A. **(G)** Representative current clamp traces of control (left, black) and ATRX miKO (right, red) firing during 400 pA (top), 200 pA (middle), and 0 pA current steps from a holding potential of −70 mV. The step protocol is shown in the bottom left. **(H)** Average spike number (CTL $n=9$, ATRX miKO $n=7$, two-way ANOVA, $F(1.000, 8.000) = 0.07102$, $p=0.796$). The solid line represents the current injection curve, and the dotted lines represent SEM. **(I)** Latency to the first spike in the rheobase sweep plotted by the current step (two-way ANOVA, $F(1, 8) = 2.608$, $p=0.145$). The dotted lines represent SEM. **(J)** Representative traces of rheobase spike in control and ATRX miKO neurons. **(K–M)** Summary of action potential amplitude (Student $T$ test, $p=0.009$), rheobase current (Student $T$ test, $p=0.1212$), and threshold (Student $T$ test, $p=0.068$). **(N)** Representative voltage clamp traces showing sEPSCs with 100 μM picrotoxin to block GABA-sIPSCs. −70 mV holding potential. **(O)** Summary of sEPSC frequency (Student $T$ test, sEPSC frequency $p=0.8277$). **(P, Q)** Cumulative distribution of sEPSC amplitude (Kolmogorov–Smirnov test, $p<0.0001$) and summary of sEPSC amplitude (Student $T$ test, sEPSC amplitude $p=0.2100$). Error bars represent SEM. The data underlying this figure can be found in the S1 Data file.

(Figs 6K and S6D). In summary, behavioral testing shows that mice lacking ATRX in microglia exhibit lower anxiety levels and impaired long-term spatial memory.

## Discussion

In this study, we find that targeting *Atrx* deletion in microglia increases chromatin accessibility, DNA damage and de-represses retroelements, triggering viral mimicry pathways and an interferon-mediated immune response. Functionally, we demonstrate that microglial-specific ATRX inactivation alters the morphology and electrophysiological properties of hippocampal CA1 neurons and impairs long-term spatial memory.

Our findings demonstrate that ATRX-mediated heterochromatin formation is required to fully suppress retroelements in microglia. The ATAC-seq data exposes a marked global de-condensation of chromatin, especially at gene TSS and retrotransposons, concomitant with aberrant transcriptional activation. Retroelements are also de-repressed in a variety of neurodegenerative, neurological, and psychiatric disorders [53–56], and ERV activation has been linked to an immune response, nucleic acid sensing response, immuno-inflammation, structural changes in hippocampal pyramidal neurons, and cognitive deficits [57,58]. Specific repression of ERVs has been shown to rescue cognitive deficits [58], suggesting that retroelement activation may contribute to neuroinflammation and cognitive deficits in ATRX miKO mice [57,58]. The deletion of ATRX in oligodendrocytes or astrocytes does not cause such extensive de-repression of retroelements as observed in microglia (S3 Fig), suggesting that epigenetic regulation of retroelements in microglia might be different and more susceptible to ATRX loss. Several layers of suppression have been described to keep retroelements in check, including DNA methylation, H3K9me3, and H3K27me3 [59,60]. ATRX can promote both H3K27me3 and H3K9me3 based on reports of its interaction with EZH2 and SETDB1, respectively [61]. Future studies could address whether microglia have relatively low DNA methylation at retroelements, which would provide an explanation for the enhanced vulnerability of ATRX-null microglia.

Our findings show that ATRX is required cell-autonomously in microglia to maintain homeostasis. In addition to larger and more rod-shaped somas and nuclei, ATRX-null microglia have hyper-ramified processes with increased complexity of branching towards the periphery. This resembles IFN-α-activated microglia [30] and challenges the notion that all activation states reduce branching. The absence of ATRX also caused genomic stability in microglia. Genomic instability drives several human diseases, such as cancer, neurodegeneration, and early aging [62–64]. We previously reported that loss of ATRX in mouse neuroprogenitor cells leads to DNA replication stress, DNA damage, and mitotic defects [65,66]. It is therefore not that surprising to detect increased DNA damage in ATRX-null microglia, which could be exacerbated by the increased proliferation of reactive microglia detected at 2 months of age. DNA damage could potentially generate cytoplasmic DNA species and trigger the DNA sensing pathway [67,68]. In addition to DNA damage, we detected bi-directional transcription of ERVs as well as non-LTR retrotransposons that can generate immunogenic small dsRNAs and R-loops [41,42,69]. These can be recognized by PRRs in the cytoplasm, such as cGAS or RIG-1, and trigger interferon-mediated

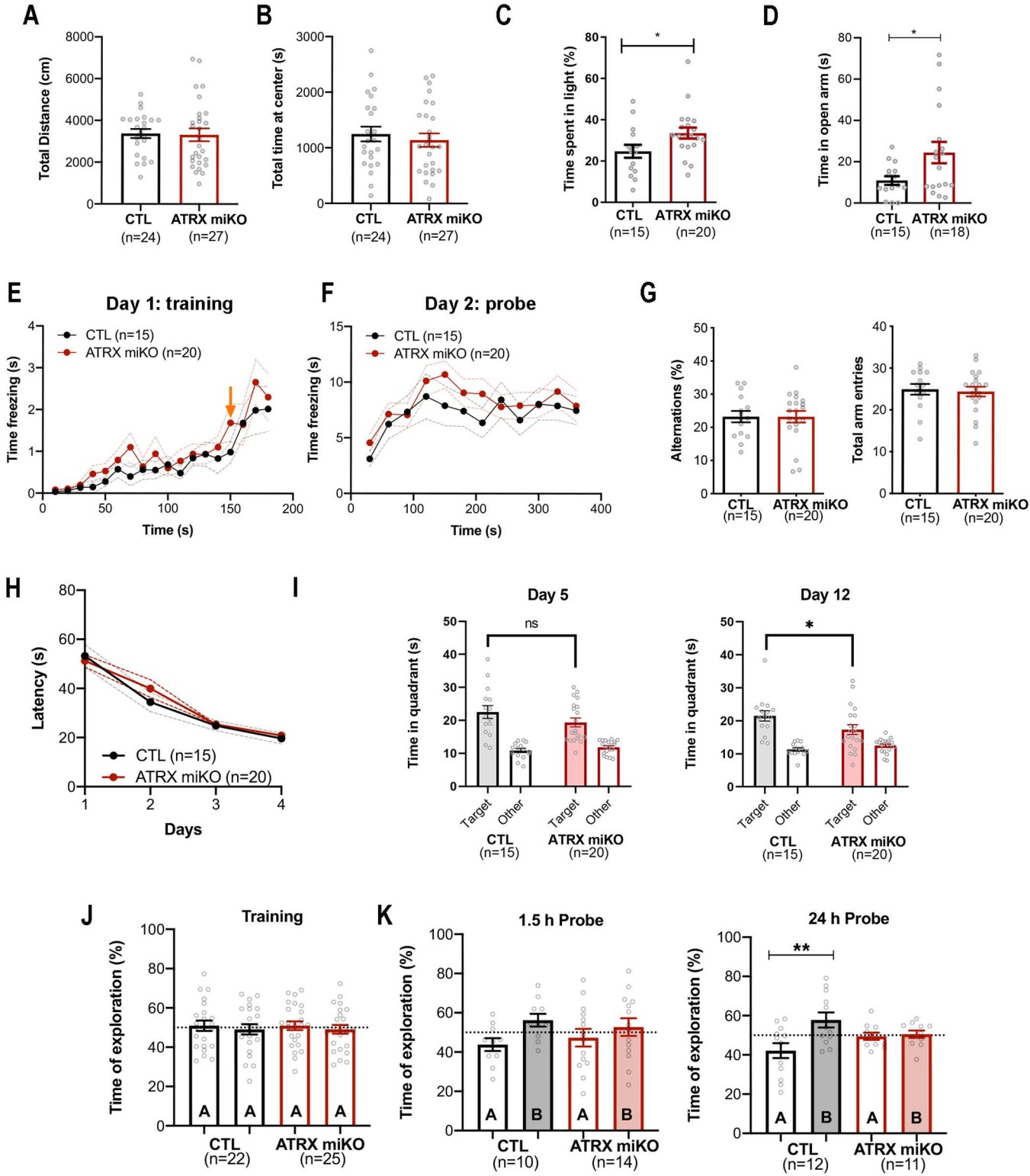

**Fig 6. Spatial and object recognition memory impairments in ATRX miKO mice. (A)** Total distance traveled in the open field test ($p = 0.884$, Student $T$ test). **(B)** Total time spent in the center of the open field ($p = 0.553$, Student $T$ test). **(C)** Percent time spent in the light chamber during the light-dark box test ($p = 0.039$, Student $T$ test). **(D)** Percent time spent in the open arm of the elevated plus maze test ($p = 0.032$, Student $T$ test). **(E)** Time freezing over

180 s during the training phase of the contextual fear conditioning task. Arrow indicates the shock administered at 150 s. **(F)** Time freezing over 360 s during the contextual fear conditioning probe test (m.c. $p = 0.416$, two-way ANOVA). **(G)** Percent spontaneous alternations and number of intersections over 5 min in the Y-maze task (percent alternations $p = 0.982$, total arm entries $p = 0.761$, Student $T$ test). **(H)** Latency to reach the platform over 4 days of training (4 trials/day) in the Morris water maze paradigm (m.c. $p = 0.5136$, two-way ANOVA). **(I)** Time spent in the target quadrant and the average time spent in the other quadrants after the removal of the platform on probe day 5 (m.c. $p = 0.1342$, two-way ANOVA), and at probe day 12 ($*p < 0.05$, two-way ANOVA). **(J)** Percent of time spent exploring object A during the 10-min training session in the novel object paradigm (m.c. CTL $p = 0.8307$, ATRX miKO $p = 0.8162$, two-way ANOVA). **(K)** Percent of time spent exploring object A and a new object B at 1.5 h (m.c. CTL $p = 0.1188$, ATRX miKO $p = 0.5472$, two-way ANOVA), and 24 h after training (CTL $p = 0.0013$, ATRX miKO $p = 0.9645$, two-way ANOVA). The dotted lines indicate random chance. Number of mice used for each test shown for every panel, m.c. = two-way ANOVA with Sidak's post-hoc test for multiple comparisons. Error bars represent ± SEM. The data underlying this figure can be found in the S1 Data file.

immune responses [43,44]. We indeed demonstrate the presence of ERV-derived dsRNAs in the cytoplasm coupled with upregulation of RIG-1, as well as downstream effectors of both cGAS and RIG1, including proinflammatory cytokines and chemokines and upregulation of ISGs. In combination, this provides substantial evidence that ATRX-null microglia undergo immune activation resulting from viral mimicry.

Although retrotransposons constitute roughly 40% of both mouse and human genomes, their evolutionary histories and functional impacts diverge in important ways. For instance, LTR retrotransposons are notably more active in mice [70]. This heightened retrotranspositional activity in mice can accelerate genomic changes and amplify the consequences of retroelement de-repression, such as increased risk of new insertions, expression of retroviral proteins, and direct stimulation of innate immune pathways [71]. In contrast, human ERVs are highly mutated and lack retrotranspositional capacity in vivo, which likely limits their direct immunogenic potential compared to their murine counterpart [72]. These differences suggest that the mechanisms and outcomes of TE de-repression and the resulting neuroinflammatory responses may be more pronounced or distinct in mice than in humans. Nonetheless, the fundamental principle that TE de-repression can activate innate immune responses is conserved: both mouse and human cells are capable of mounting robust antiviral responses upon sensing cytosolic nucleic acids derived from retroelements, implicating viral mimicry as a shared pathogenic mechanism [73].

Our findings also highlight important non-cell autonomous consequences of ATRX deletion in microglia on hippocampal neurons, such as significantly increased branching of CA1 dendrites and increased density of filopodia spines on mid-apical dendrites. This could potentially be explained by hyper-ramification of ATRX-null microglia in this brain region, resulting in increased microglia-dendrite interactions and inducing filopodia formation [74]. Alternatively, microglia lacking ATRX might fail to eliminate "weaker signals" such as filopodia spines [75].

Neurophysiological assessment of CA1 pyramidal neurons detected a decrease in the input resistance. The lower input resistance generally indicates hypo-excitability of the neurons because, by Ohm's law ($V = I \times R$), a given positive ion influx ($I$) by excitatory synaptic transmission results in a smaller postsynaptic depolarization ($V$) with lower input resistance ($R$). On the other hand, microglial ATRX deficiency also led to an increase in the action potential amplitude. While the classic view is that action potential transmits signals in all-or-none fashion [52], a growing body of evidence shows that changes in the action potential amplitude can fine-tune the $Ca^{2+}$ dynamics at the axon terminals and the ensuing neurotransmitter release [76–78]. Thus, our data suggests that microglial ATRX deficiency may change signal transmission from CA1 pyramidal neurons to their downstream targets. Mechanistically, an increase in action potential amplitude can be explained by an increase in the voltage-gated Na+ channels that drives depolarization and/or a decrease in voltage-gated K+ channels that counteract the depolarization. Future studies on the precise biophysical changes in CA1 pyramidal neurons will be required to clarify the mechanisms by which microglia influence neuronal morphology and function.

The behavior tests indicate that ATRX deletion in microglia leads to selective impairment of long-term spatial memory, as assessed in the Morris water maze task. This deficit was highly specific, as recognition memory, contextual

fear conditioning, and working memory remained largely intact. These findings argue against a broad cognitive role for ATRX-dependent pathways in microglia. Instead, they support a model where microglial ATRX is essential for maintaining chromatin and immune homeostasis, and its loss has a precise, downstream impact on spatial memory consolidation.

It is not yet clear which of the identified alterations in microglia is responsible for neuronal changes displayed by ATRX miKO mice. Loss of microglia homeostasis can result in the release of proinflammatory cytokines and chemokines that could in theory influence synaptic plasticity and cognitive processes [79–84]. Indeed, overproduction of these cytokines and chemokines has previously been demonstrated to modulate neuronal activity and cause cognitive deficits [83,85–90]. There is limited knowledge on how cytokine production can affect neuronal excitability, although there is evidence that it leads to upregulation of sodium current densities in neurons [91]. Alternatively, cytokines can directly interact with neuronal receptors. For example, IL-1 may increase neuronal excitability through direct interaction of its receptor complex with NMDA receptors [92]. Alternatively, cytokines could induce transcriptional changes in neurons, altering the expression of ion channels and receptors that regulate excitability. Microglia function as the central hubs for innate immune signaling in the brain, initiating a cascade that extends to other cell types [93]. The activation of RIG-I by endogenous dsRNAs in microglia leads to the secretion of type I interferons, which act as potent paracrine signals. These IFNs can engage receptors on nearby astrocytes and neurons, enhancing their antiviral capabilities by activating their own RIG-I pathways. This intercellular signaling network may allow ATRX-null microglia to amplify antiviral and inflammatory responses across different cell types in the brain, creating a more coordinated defense against potentially harmful endogenous RNA species.

A key implication of our work is that neuronal changes and memory consolidation deficits in ATR-X syndrome may be treatable by targeting the underlying microglial reactivity. Our model provides a strong rationale for exploring several immunomodulatory strategies. These include the targeted repression of endogenous retroelements [58], inhibition of damaging nucleic acid-sensing pathways [57], neutralization of specific inflammatory cytokines [94], or the direct suppression of detrimental microglial immune activation [10,13,14]. By shifting the focus from neuron-intrinsic defects to glia-driven pathology, this work establishes a new guiding theory for therapeutic intervention in ATR-X syndrome.

## Materials and methods

### Animal husbandry and genotyping

Mice were exposed to the 12-h light/12-h dark cycles and fed regular chow and water ad libitum. ATRX$^{loxP}$ mice were previously described [16], and two reporter lines (Sun1GFP (B6;129-Gt(ROSA)26Sor$^{tm5(CAG-Sun1/sfGFP)Nat}$/J, MGI:5614796, RRID: IMSR_JAX:021039) and Tomato-Ai14 (Ai14) allele (B6.Cg-Gt(ROSA)26Sor$^{tm14(CAG-tdTomato)Hze}$/J, MGI:3809524, RRID:IMSR_JAX:007914)) were bred with the ATRX$^{loxP}$ mice [27,28]. Upon Cre-mediated recombination, Sun1GFP green fluorescent protein will label the nuclear membrane, while Tomato-Ai14 will express tdTomato protein with red fluorescence in the cytoplasm and nucleus. Male progeny with *Atrx* deficiency in microglia upon tamoxifen treatment was produced by mating ATRX$^{loxP}$;Sun1GFP or ATRX$^{loxP}$;Tomato-Ai14 females to males expressing Cre under the inducible promoter of Cx3Cr1$^{ER}$ (B6.129P2(Cg)-Cx3cr1$^{tm2.1(cre/ERT2)Litt}$/WganJ, MGI:5617710, RRID:ISMR JAX:021160) [26]. To delete ATRX in astrocytes, C57BL/6 heterozygous *Atrx*$^{loxP}$ females were mated with heterozygous B6 Glast-CreER (Tg(Slc1a3-cre/ERT)1Nat, IMSR JAX:012586, MGI:4430111) male mice that express inducible Cre recombinase under the control of the *Slc1a3* (Glast) promoter [95]. Mating *Atrx*$^{loxP}$ female mice to males expressing Cre recombinase under the control of the inducible Sox10 promoter [96] (CBA;B6-Tg(Sox10-icre/ERT2)388Wdr, MGI:5634390, RRID:IMSR JAX:027651), produced male progeny with *Atrx* deficiency in OPCs (*Atrx*$^{loxP}$;Sox10Cre or *Atrx*$^{Sox10Cre}$) upon tamoxifen treatment, as described by Rowland and colleagues [97]. Genomic DNA was extracted from an ear notch and genotyped as described previously [32]. The genotyping primers are listed in S11 Table. Mouse weight was checked with weighing balance at 2 and 3 months of age. Behavioral tests were performed using male mice of 3–6 months of age, starting from less demanding to more demanding ones (open field tests, light-dark box, elevated plus maze, Y-maze, novel object recognition, fear conditioning, and Morris water maze). All behavioral tests were performed on at least 15 animals per genotype between 9:00 AM and 4:00 PM.

ARRIVE guidelines were followed: mouse groups were randomized, experimenters were blind to the genotypes, and software-based analysis was used to score mouse performance in all the tasks. The Animal Care and Use Committee of the University of Western Ontario approved all animal procedures in compliance with the Animals for Research Act guidelines of the province of Ontario, Canada (AUP-2021-049 and AUP-2021-064).

## Tamoxifen administration

Tamoxifen (10 mg; Cat#T5648, Sigma) was dissolved in 100 µl 95% ethanol at 65 °C for 10 min and then diluted with 900 µl corn oil (Cat# C8267, Sigma). 45-day-old adolescent male mice were injected intraperitoneally daily with 2 mg tamoxifen for 5 consecutive days.

## Immunofluorescence

Two- or three-month-old mice were *trans*-cardially perfused and fixed as described previously [97]. Fixed brains were then sectioned coronally at 8–10 µm thickness (Leica CM 3050S) on Superfrost slides (Thermo Fisher Cat# 22-037-246) and stored at −80 °C with a desiccant (VWR, 61161-319). For the immunofluorescence, slides were rehydrated in 1×PBS for 5 min followed by washing with wash buffer (1×PBS + 0.3% TritonX-100 (Millipore Sigma Cat# T8787)). Antigen retrieval was performed (except for CD68 staining) by incubating slides in 10 mM sodium citrate pH6 at 95 °C for 10 min. Cooled sections were washed in 1×PBS and then blocked with 5% goat serum (Millipore Sigma Cat# G9023) or donkey serum (Millipore Sigma Cat# D9663) in wash buffer and incubated with primary antibody overnight at 4 °C. After washing three times (5 min each) with wash buffer, slides were incubated with secondary antibody for 1 h at room temperature in the dark. Slides were washed twice for 5 min with washing buffer, and counterstained with 1 µg/mL DAPI (Millipore Sigma Cat# D9542) for 5 min followed by 1× PBS wash. Finally, sections were mounted with Permafluor (Thermo Fisher Cat# TA-006-FM) and imaged with an inverted microscope (DMI 6000b, Leica) equipped with a digital camera (ORCA-ER, Hamamatsu). Volocity (PerkinElmer Demo Version 6.0.1, RRID:SCR_002668) and Adobe Photoshop was used for image processing. All cell counts were performed in a blinded and randomized manner. Primary antibodies used for immunofluorescence were anti-ATRX (Santa Cruz Biotechnology, Cat# sc-15408, RRID:AB_2061023), anti-CD68 (Biorad, Cat#MCA1957, RRID:AB_322219), anti-Ki67 (abcam, Cat# ab15580, RRID:AB_443209), anti-IBA1 (Cedarlane, Cat# 019-19741, RRID:AB_839504), anti-IBA1 (Novus Biologicals, Cat# NB100-1028, RRID:AB_3148646), anti-γH2AX (Cell Signaling Technology Cat# 2577, RRID:AB_2118010), anti-GFP (Thermo Fisher Scientific, Cat# PA1-9533, RRID:AB_1074893), anti-DCX (NEB cat# 4604S), anti-NeuN (Millipore Sigma cat# MAB377), and anti-activated Caspase 3 (NEB cat# 9661S). The following secondary antibodies were used: goat anti-rabbit-Alexa Fluor 594 (1:800, Thermo Fisher Scientific, A-11012, RRID:AB_2534079), goat anti-rabbit-Alexa Fluor 488 (1:800, Thermo Fisher Scientific Cat# A-11008, RRID:AB_143165), donkey anti-sheep-Alexa Fluor 594 (1:800 Thermo Fisher Scientific Cat# A-11016, RRID: AB_2534083), goat anti-chicken-Alexa Fluor 488 (1:800, Thermo Fisher Scientific Cat# A-11039, RRID:AB_2534096), donkey anti-goat-Alexa Fluor 594 (1:500 Thermo Fisher Scientific Cat# A-11058, RRID:AB_2534105) and goat anti-rat-Alexa Fluor 488 (1:800, Thermo Fisher Scientific, A-11006, RRID:AB_2534074). Slides were washed twice for 5 min with 0.3% Triton X 1×PBS, followed by DAPI staining for 5 min, washed, and mounted with Permafluor Aqueous Mounting Medium (Fisher Scientific cat#TA-030-FM).

## Analysis of microglia cell morphology

Tomato-Ai14 (Ai14) reporter mice (B6.Cg-Gt(ROSA)26Sor^tm14(CAG-tdTomato)Hze/J, MGI:3809524, RRID:IMSR_JAX:007914)) were bred with the ATRX^loxP mice to enable induction of tdTomato protein in the cytoplasm and nucleus in a Cre-dependent manner. Fixed brains were sectioned at 10 µm thickness using a cryostat or 50 µm thickness on a Vibratome Series 1000. Cryosections were counterstained with 1 µg/mL DAPI (Millipore Sigma Cat# D9542) for 5 min, washed in 1×PBS, mounted with Permafluor (Thermo Fisher Cat# TA-006-FM) and imaged with an inverted microscope (DMI 6000b, Leica) equipped with a digital camera (ORCA-ER, Hamamatsu). Volocity (PerkinElmer Demo Version 6.0.1, RRID:SCR_002668)

and Adobe Photoshop was used for image processing. Microglia (Ai14+) were classified based on the shape of the soma and nuclei, i.e., round, elongated, and medium. Thicker sections were mounted and imaged on a laser scanning confocal microscope (Leica Stellaris 5). Z-stacks of 0.5 μm thickness were obtained of the hippocampal CA1 region. Projections were traced and analyzed in a blinded manner using Neurolucida 360 Studio (MBF Bioscience, V. 2023.1.1) and Neurolucida Explorer (MBF Bioscience, V. 2022.2.1), respectively. Convex Hull 3D was used to determine volume. In this analysis, a convex polygon is generated by connecting the tips of the distal branches of microglia and the volume of the polygon is reported. The maximum order of branching for each microglia was determined and the average number of microglia for each maximum branching order was arranged in bins and plotted. Data was obtained from 8 to 15 microglia per animal from 3 mice of each genotype.

### Fluorescence-activated sorting of microglia nuclei

FANS was performed as described previously [32]. The cortex and hippocampus of 2-month-old mice were homogenized in 20 mM Tricine KOH, 25 mM $MgCl_2$, 250 mM sucrose, 1 mM DTT, 0.15 mM spermine, 0.5 mM spermidine, 0.1% IGEPAL-630, 1× protease inhibitor cocktail (Millipore Sigma Cat# 11873580001), 1 μL/mL RNase inhibitor (Thermo Fisher Scientific Cat# 10777019)). After dilution with homogenization buffer and filtering through a 40 μm strainer (Fisherbrand Cat#22363547), the samples were layered on the top of 1:1 volume cushion buffer (0.5 mM $MgCl_2$, 0.88 M sucrose, 0.5 mM DTT, 1× protease Inhibitor cocktail (Millipore Sigma Cat# 11873580001), 1 μL/mL RNase inhibitor (Thermo Fisher Scientific Cat# 10777019). Nuclei were then pelleted at 2,800$g$ for 20 min at 4 °C and incubated for 10 min in sorting buffer (4% FBS, 0.15 mM spermine, 0.5 mM spermidine, 1× protease inhibitor cocktail (Millipore Sigma Cat# 11873580001) and 1 μL/mL RNase inhibitor (Thermo Fisher Scientific Cat# 10777019) in 1×PBS). After resuspending and filtering through a 20 μm strainer (PluriSelect Cat#431002060), nuclei were sorted using a Sony SH800 Cell sorter and Sun1GFP+ microglia nuclei were collected into a 1.5 ml tube.

### RNA purification and RNA-seq library preparation

RNA extraction followed by RNA-seq library preparation was performed as described previously [32]. Briefly, sorted Sun1GFP+ microglia nuclei were collected directly in lysis buffer (supplemented with 2% β-mercaptoethanol) from the single cell RNA purification kit (NorgenBiotek Cat#51800). RNA extraction, with on-column DNase treatment, was performed by following the manufacturer's instructions. 35 ng of total RNA was used to deplete rRNA (Ribo-off rRNA Depletion kit (H/R/M), Vazyme Cat#N406), followed by strand-specific RNA-seq library preparation using the VAHTS Universal V8 RNA-seq Library Prep Kit for Illumina (Vazyme Cat#NR605-01).

### RNA-seq analysis

RNA-seq libraries were sequenced at Canada's Michael Smith Genome Sciences Centre (BC Cancer Research, Vancouver, BC, Canada) using the Illumina HiseqX (Illumina, San Diego, CA), and 60−120 million paired-end reads (150 bp) were obtained for each library. RNA-sequencing analysis was performed as described previously [32,97]. Briefly, Trim galore v0.6.6 with the following parameters ('–phred33 -length 36 -q 5 -stringency 1 -e 0.1') was used to trim the raw data, and HISAT2 version 2.0.4 [98] mapped the paired-end reads against the *Mus musculus* GRCm38.p6 (primary assembly downloaded from Ensembl). SAMtools [99] was then used to sort and convert SAM files. StringTie v.2.1.5 [100] was used to obtain gene and transcript abundance for each sample by providing read alignments and *Mus musculus* GRCm38 genome annotation as input. Tximport R/Bioconductor package was used to import transcript coverage and abundance into R, and the DESeq2 R/Bioconductor package [101] conducted a differential analysis of transcript count data. The independent hypothesis weighting method was used to weigh $P$-values and adjust for multiple testing using the procedure of Benjamini–Hochberg (BH). Finally, the Lancaster method was used to aggregate transcripts' $p$-values. Gene ontology and gene cluster analysis were performed as described previously [97]. For GSEA, significant DEGs were ranked

based on log2FoldChange. gseGO function from clusterProfiler v4.0.5 [102] was then used with the following parameters (ont = "BP", OrgDb = org.Mm.e.g.,db, minGSSize = 10, maxGSSize = 500, eps = 1e-10, pvalueCutoff = 0.05, pAdjustMethod = "BH") to perform GSEA. For single-cell deconvolution, reference-based decomposition mode from the R toolkit, Bisque v1.0.5 [103], was used to estimate cell composition from our bulk expression data with microglia single-cell data downloaded from GSE142267 [33].

## Assay for transposase-accessible chromatin (ATAC)

ATAC-seq on sorted nuclei was performed as described previously [32]. Briefly, sorted Sun1GFP+ microglia nuclei were collected directly in 1× PBS and pelleted at 3,000 rpm for 5 min at 4 °C. The nuclei were resuspended directly in tagmentation buffer containing Tn5 transposase (Vazyme Cat#TD501) and the reaction mixture was placed at 37 °C for 45 min. After tagmentation, samples were purified using the QIAquick PCR Purification Kit (Qiagen Cat#28104) and amplified by PCR according to the TruePrep DNA kit instructions (Vazyme Cat#TD501). Quantitative PCR was performed to determine the optimal number of cycles (1/3 saturation), and the libraries were amplified with no more than 12 PCR cycles.

## ATAC-seq analysis

On average, 100M paired-end reads (150 bp) were obtained for each library. Raw reads were trimmed with Trim Galore (v.0.6.6) with the following parameters ('–phred33 -length 36 -q 5 -stringency 1 -e 0.1') and mapped to *Mus musculus* GRCm38.p6 using Bowtie2 v2.4.4 [104] with the default parameters. SAMtools v1.12 was then used to create and sort BAM files from the aligned reads recorded in SAM format. The duplicated reads were then marked with the MarkDuplicates function from Picard (v.2.26.3). The mitochondrial DNA reads, and blacklist regions of the genome were filtered out using Bedtools intersect (v.2.30.0). Two approaches were employed to identify differentially accessible regions (DARs). First, MACS2 v2.2.7.1 [105] (with parameters --nomodel -f BAMPE) was used to call the peaks for each sample. The peaks from all samples were merged into a set of non-redundant open regions with Bioconductor package soGGi v1.24.1. The peaks that were present in at least two samples were kept for downstream analysis. The featurecounts function from Rsubread v2.6.4 [106] was used to count the reads from each sample overlapping the consensus peak set and DESeq2 v1.32.0 was used to perform a differential analysis of accessible regions between ATRX miKO and control samples. Second, R package csaw (v.1.26.0) [107] was used to count the reads in 200 bp non-overlapping windows. Background noise was estimated by counting reads in 2,000 bp bins. We selected 200 bp windows that have a signal higher than log2(3) above the background. Windows were then merged if less than 100 bp apart but did not extend above 5kb width. EdgeR 3.40.2 [108] was then used to identify the DARs. Because MACS2 and csaw have more than 93% overlap in calling DARs (S2B Fig), only MACS2 was later used for downstream analysis. DARs (*P*-value < 0.05) in the genome were annotated using ChIPseeker v1.28.3 [109]. For TF footprinting, TF motifs were downloaded from the JASPAR CORE database [110]. The footprinting analysis was performed using merged BAM files of each condition and a merged set of peaks from all samples. We used TOBIAS v0.12.11 [34] to assess chromatin occupancy by TFs. TOBIAS ATACorrect was used to correct Tn5 insertion bias in input BAM files. Footprinting scores were then calculated using TOBIAS ScoreBigWig. We then used TOBIAS BINDetect to analyze the differential binding of TFs between ATRX miKO and control groups.

## Intersection of RNA-seq and ATAC-seq

To generate heatmaps of expression levels, gene-level counts from the Stringtie results were transformed using the variance stabilizing transformation (VST) method in DESeq2. For the heatmap of chromatin accessibility, ATAC-seq signals at promoters of the genes (1,000 bp upstream and 1,000 bp downstream of TSS) were counted using the featureCounts function from Rsubread v2.6.4. Z-scores were calculated for expression or accessibility independently. Heatmaps were plotted using the Pheatmap package.

## Analysis of repetitive elements

For the analysis of repetitive elements, we first took a broad, per locus approach. The repeatmasker annotation file for mm10 was downloaded from the UCSC table browser. Reads falling within the repeatmasker annotation were counted for both RNA and ATAC assays using the featureCounts function from Rsubread v2.6.4. For both RNA-seq and ATAC-seq, reads were aligned using HISAT2 v2.0.4 (RNA-seq) or Bowtie2 v2.4.4 (ATAC-seq) with default parameters. For quantification with featureCounts, only uniquely mapping, non-duplicate reads were counted (countMultiMappingReads = FALSE; --ignoreDup). The average rate of uniquely mapped reads assigned to the repeatmasker annotation was approximately 25% for both RNA-seq and ATAC-seq across all samples (control and miKO). Full mapping details, including the number of assigned, duplicate, no feature, ambiguous, and unmapped reads for each sample, are provided in S13 Table. Differential expression and accessibility of repeats were determined using DESeq2. For the bi-directional transcription of retroelements, mapped reads from RNA-seq data in BAM format were split into sense and antisense strand BAMs using a custom script. Reads falling within the UCSC repeatmasker annotation were counted for both strands separately using the featurecounts function from Rsubread. Bigwig tracks were generated using Deeptools11 bamCoverage v.3.5.2 with the parameters "-bs 25 –normalize using RPKM" and visualized in the UCSC genome browser. In a second approach, we performed TE expression analysis using TEtranscripts (version 2.0.3) [111], a tool specifically designed for accurate TE quantification. We utilized the curated GTF file provided by the TEtranscripts authors, which excludes simple repeats and various RNA species (e.g., rRNAs, scRNAs, snRNAs, srpRNAs, and tRNAs), ensuring the analysis focused solely on transposable elements. RNA-seq reads were aligned to the GRCm38/mm10 reference genome using STAR aligner (version 2.7.3a) [112] with parameters optimized for TE analysis:

- --winAnchorMultimapNmax 100 to allow for multiple mappings of repetitive sequences.

- --outFilterMultimapNmax 100 to include reads mapping up to 100 locations.

TEtranscripts was run in multi-mode (--mode multi) to accurately assign multi-mapping reads, which is essential due to the repetitive nature of TEs. To assess bidirectional transcription, mapped reads from RNA-seq data in BAM format were split into sense and antisense strand BAMs using a custom script. Reads falling within the UCSC repeatmasker annotation were counted for both strands separately using featurecounts function from Rsubread. Differential expression analysis of repeats for forward and reverse strands was performed using DESeq2.

## Identification and classification of upregulated LTR elements

We began with a list of TEs showing significant differential expression in our RNA-seq analysis. From this set, we selected the subset of LTR elements with positive log2 fold change (log2FC > 0), resulting in 3,202 upregulated LTR loci. For each of these loci, genomic coordinates were extended by ±15 kilobases (kb) relative to the reference mouse genome (mm10) to provide sufficient flanking sequence for identification of complete endogenous retrovirus (ERV) structures. Genomic sequences for the extended regions were retrieved using the BSgenome.Mmusculus.UCSC.mm10 package in R, and individual FASTA files were generated for each locus. These FASTA files were then analyzed using RetroTector [113], which was run in batch mode (one sequence per subdirectory) using the SweepDNA and SweepScript executors with default parameters. RetroTector classified each locus based on the presence and completeness of canonical retroviral motifs. Elements containing both flanking LTRs and all major internal domains were classified as full-length proviruses. Elements missing one or more of these internal domains, but with evidence of partial internal structure, were classified as partial proviruses. LTRs lacking substantial internal retroviral motifs were categorized as solo LTRs. Classification results were parsed from RetroTector chain files using a custom R script, which extracted annotation and structural information and merged the output with the original LTR annotation table.

## Mixed glial primary culture

Cre recombination in pups was induced by IP injection of lactating mothers from postnatal days 1 to 3 (P1–P3). At P4, cortices were used to establish mixed glial cultures as previously described [97] with some modifications. Briefly, tissue was dissociated and incubated at 37 °C for 20 min in papain solution (Worthington Cat# LS003124) supplemented with Turbo DNase (AM2238). Serum-supplemented media was added to deactivate papain followed by a 10 min incubation at room temperature and trituration with a flame-polished Pasteur pipette until a homogeneous suspension was obtained. Samples were centrifuged at 300 $g$ for 5 min, pellets were resuspended in 2 ml of DMEM supplemented with 10% FBS and Primocin (10 mg/ml, InvivoGen, Cat# ant-pm-1). Cells were added to Millicell EZSlide (Millipore, Cat#PEZGS0816) wells coated with poly-L-lysine (1 mg/ml) and incubated in supplemented media in an incubator at 35 °C and 5% $CO_2$ for 3 h to allow cells to attach before a full media change. Media was changed after 48 h and 5 days in vitro (DIV). At 7DIV, cells were fixed with 4% PFA at room temperature for 15 min, permeabilized with 0.25% Triton-X in RNase-free PBS, washed once with 0.05% tween 20 in PBS and blocked for 1 h at room temperature with 3% BSA in PBS. Cells were incubated with J2 antibody (1:200, SCICONS # 10010200) for 1 h, followed by two washes and incubation with goat anti-mouse-Alexa Fluor 488 (1:1000, Thermo Fisher Scientific Cat# A-11001, RRID:AB_2534069) for 1 h at room temp to detect dsRNA. The wells were counter-stained with DAPI before mounting.

## dsRNA immunoprecipitation

Control and ATRX miKO forebrain and cerebellum were harvested, minced, and mixed with 1.2 ml cold lysis buffer (10 mM Tris pH7.0, 10 mM NaCl, 5 mM $MgCl_2$, 0.5% IGEPAL CA-630, 0.5% Triton X-100, 10U/ml DNAse I, Sigma, and 40 U/µl RNaseOut inhibitor, Thermo Fisher). The lysate was centrifuged at 3,000$g$ for 3 min at 4 °C, and the supernatant transferred to a fresh tube and further centrifuged at 21,000$g$ for 5 min at 4 °C. The supernatant (cytoplasmic fraction) was collected for RNA IP and incubated with 5 µl anti-dsRNA monoclonal antibody J2 (SCICONS # 10010200, Mouse/ IgG2a, Kappa, Thermo Fisher) or Mouse IgG2a, Kappa Monoclonal [MOPC-173]-isotype control (ab18413, Abcam) for 2 h at 4 °C. Protein G-Dynabeads (10003D, Thermo Fisher) were washed twice using RIP buffer (50 mM Tris, pH7.4, 100 mM NaCL2, 3 mM MgCl and 0.5% IGEPAL CA-630), and incubated with the sample-antibody mix for another hour at 4 °C. The dsRNA-antibody-Dynabeads complex was washed four times with RIP buffer. RNA was extracted using 1 ml Trizol and 200 µl chloroform. After isopropanol precipitation and washing with 75% ethanol, RNA was resuspended in 25 µl RNase-free water cDNA was synthesized by using previous Reverse Transcriptase and protocol [17]. PCR primers used were as follows: ERVB2_1-I_MM-intDup 5, Forward 5′-AGCACATGTTGTCCAATCGG-3′, Reverse 5′-CTCTCCTC CAGAAACCAGGG-3′ (102 bp amplicon). ERVL-MaLR-MLT1C, Forward 5′-TGAGGATGGTGAGCTGAGTT-3′, Reverse 5′-TGCTCTCCACAGTGTCCATT-3′(124 bp amplicon).

## Western blot analysis

Total protein was extracted from the cortex and hippocampus of 2-month-old mice using the RIPA buffer (150 mM NaCl, 1% NP-40, 50 mM Tris pH 8.0, 0.5% deoxycholic acid, 0.1% SDS, 0.2 mM PMSF, 0.5 mM NaF, 0.1 mM Na3VO4, 1× protease inhibitor cocktail (Millipore Sigma Cat# 11873580001). Homogenized samples were incubated on ice for 30 min and centrifuged at 12,000 rpm for 15 min at 4 °C. Bradford assay (BioRad Cat# 500–0006) was used to quantify the protein and 100 µg of protein was resolved on 10% SDS-PAGE gel. After transferring proteins to nitrocellulose membrane (BioRad Cat# 1620115), the membrane was blocked with milk-TBST (5% skimmed milk, 1×TBS and 0.1% Tween-20) and then incubated with anti-STAT1 (Cell Signaling Technology Cat# 9172, RRID:AB_2198300), anti-pSTAT1 (Cell Signaling Technology Cat# 9167, RRID:AB_561284), anti-RIG-1 (Cell Signaling Technology Cat# 3743, RRID:AB_2269233), anti-cGAS (Cell Signaling Technology Cat# 31659, RRID:AB_2799008), anti-Actin (Sigma-Aldrich Cat# A2066, RRID:AB_476693) antibodies at 4 °C overnight. After three washes with milk TBST, the membrane was incubated

with the appropriate secondary antibody (rabbit anti-HRP (1:5000, Jackson ImmunoResearch Cat# 111-036-003, RRID:AB_2337942); mouse anti-HRP (1:5000, Santa Cruz Cat# sc-516102, RRID:AB_2687626)). Finally, the membrane was incubated in an enhanced chemiluminescent solution (Thermo Fisher Cat# 34095) and exposed using the Universal Hood III (BioRad Cat# 731BR00882). Quantification of blots was performed with ImageJ (version 1.53).

## Cytokine/chemokine analysis

Total protein was extracted from the cortex and hippocampus of 2-month-old mice using the RIPA buffer and the multiplexing analysis was performed using the Luminex 200 system (Luminex, Austin, TX, USA) by Eve Technologies Corp. (Calgary, Alberta). Forty-five markers were simultaneously measured in the samples using Eve Technologies' Mouse Cytokine 45-Plex Discovery Assay which consists of two separate kits; one 32-plex and one 13-plex (MilliporeSigma, Burlington, Massachusetts, USA). The assay was performed according to the manufacturer's protocol. Assay sensitivities of these markers range from 0.3 to 30.6 pg/mL for the 45-plex.

## Analysis of CA1 neurons and dentate gyrus

The Thy1-GFP-M allele was introduced in the ATRX floxed line to achieve sparse neuronal labeling. Fixed brains were sectioned at 100–150 µm thickness on a Vibratome Series 1000. Sections were mounted and imaged on a laser scanning confocal microscope (Leica Stellaris 5). Z-stacks of 0.5 µm (spine analysis) or 1.5 µm (dendrite analysis) thickness were obtained of the hippocampal CA1 region. Projections were traced and analyzed in a blinded manner using Neurolucida 360 Studio (MBF Bioscience, V. 2023.1.1) and Neurolucida Explorer (MBF Bioscience, V. 2022.2.1), respectively. For dendrite analysis, 8 neurons were imaged in 5 mice per genotype. A two-way ANOVA was performed in GraphPad Prism software. For spine analysis, five separate spine segments were measured in six neurons per mouse ($n = 5$ mice per genotype). Each spine was analyzed using Neurolucida Explorer with consistent detection settings. The Student $T$ test (unpaired, two-tailed) was performed using GraphPad Prism. For DG analysis, brain sections were imaged at 10× magnification using an inverted Leica microscope (CTR 6500) using Volocity Software (version 7.0.0). Brain sections ($n = 5$–6) were quantified per brain using ImageJ for 5 mice per genotype. Ki67, DCX, and aCASP3 staining densities were normalized to the length of the inner blades of the DG.

## Slice preparation for electrophysiology

Mice were deeply anesthetized using sodium pentobarbital (100 mg/kg intraperitoneally) and transcardially perfused with cold (2–4 °C) NMDG-HEPES solution (92 mM NMDG, 93 mM HCl, 2.5 mM KCl, 1.2 mM $NaH_2PO_4$, 30 mM $NaHCO_3$, 20 mM HEPES, 25 mM Glucose, 5 mM sodium ascorbate, 2 mM Thiourea, 3 mM sodium pyruvate, 10 mM $MgCl_2$, 0.5 mM $CaCl_2$ (300–310 mOsm), saturated with 95% O2/5%$CO_2$). Brains were quickly removed and placed in a cold NMDG-HEPES solution for slicing. Coronal sections (350 µm thick) containing the hippocampus were cut using a vibratome (VT-1200, Leica Biosystems). Slices were incubated at 34 °C for 15 min in NMDG-HEPES solution saturated with 95% O2/5%$CO_2$. Slices were then transferred to artificial cerebrospinal fluid (aCSF) (126 mM NaCl, 2.5 mM KCl, 26 mM $NaHCO_3$, 2.5 mM $CaCl_2$, 1.5 mM $MgCl_2$, 1.25 mM $NaH_2PO_4$, and 10 mM D-glucose (295–300 mOsm)), saturated with 95% $O_2$/5%$CO_2$ and maintained at room temperature until recording.

## Electrophysiology measurements

Slices were transferred to a recording chamber superfused with aCSF at a flow rate of 1.5–2.0 mL/min and maintained at 27–30 °C. CA1 neurons were visualized using an upright microscope with infrared differential interference contrast optics (BX 51WI, Olympus). Borosilicate glass recording pipettes (BF120-69-15, Sutter Instruments) were pulled in a Flaming/Brown Micropipette Puller (P-1000, Sutter Instruments) with a resistance between 3 and 5 MΩ. Pipettes were filled with an

internal solution (116 mM K-gluconate, 8 mM KCl, 12 mM Na-gluconate, 10 mM HEPES, 2 mM MgCl$_2$, 4 mM K2ATP, 0.3 mM Na3GTP, and 1 mM K2-EGTA (283–289 mOsm, pH 7.2–7.4)). Spike firing was measured in the current clamp from a holding potential of −80 mV using a step protocol from −160 to 460 pA in 40 pA increments. Glutamatergic sEPSCs were isolated by adding picrotoxin (100 μM) to the aCSF while holding the postsynaptic neuron at −80 mV in a voltage clamp. Access resistance was monitored throughout the recording and cells were discarded if the value exceeded 20 MΩ.

### Data collection and analysis of electrophysiological measurements

Whole cell patch clamp recordings were obtained using a Multiclamp 700B amplifier (Molecular Devices, California, USA), low-pas filtered at 1 kHz and digitized at a sampling rate of 20 kHz using Digidata 1440A (Molecular Devices). Data was recorded on a PC using pClamp 10.6 (Molecular Devices) and analyzed using MiniAnalysis (Synaptosoft, Georgia, USA) for EPSCs, Clampfit (Molecular Devices) for membrane potential and a custom Python code (adapted from a white paper from Allen Cell Types Database: https://github.com/AllenInstitute/ipfx) for cell firing. Briefly, the slope (*dV/dt*) was measured by taking the difference in voltage between two time steps and dividing it by the resolution of acquisition. The time of the threshold crossing was detected by finding the time point where *dV/dt* was ≤5% of the maximum dV/dt of the rising phase. The following criteria were used to detect action potentials during current injection steps: (1) the duration from threshold to the peak is ≤5 ms, (2) amplitude is ≥2 mV and absolute peak ≥−30 mV, and (3) action potential trough (minimum membrane potential in the interval between the peaks of two consecutive action potentials) is ≤–22 mV. For sEPSC analysis, baseline data was taken at least 5 min after breaking through into whole-cell mode and a 0.5- or 1-min bin was used for analysis. To achieve an accurate measure of the amplitude, individual sEPSCs were visually screened in Mini-Analysis and events below 5 pA were not included.

### Open field test

The mice were acclimated to the room for 30 min prior to testing. Then the mice were placed in an open arena (length 20 cm, width 20 cm, height 30 cm) for 2 h and locomotor activity was measured in 5 min intervals as previously described [17]. Locomotor activity was automatically recorded (AccuScan Instrument), and distance traveled, and time spent in the center were reported.

### Y maze test

The mice were acclimated to the room for 30 min prior to testing. The mice were placed in the center of a symmetrical three-armed Y maze as described previously [17] and their activity was recorded for 5 min using AnyMaze. The order and number of entries into each arm were reported. Spontaneous alternations were counted when a mouse entered all three arms in a row without visiting a previous arm.

### Light dark box

The mice were acclimated to the room for 30 min prior to testing and were then placed in a light-dark box and allowed to explore the arena for 10 min. Their activity was automatically recorded (AccuScan Instrument), and the percent time spent in light was reported.

### Elevated plus maze

The mice were acclimated to the room for 30 min prior to testing and then placed in the center of the elevated plus maze (Med Associate) and their activity was recorded for 5 min using AnyMaze. Time spent in the open, closed, or center of the elevated plus maze was reported. The center of the mouse body was used as an indicator to determine its presence in an open, closed, or center.

### Novel object recognition

The mice were habituated for two consecutive days in an empty arena (40 cm × 40 cm) for 5 min as described previously [17]. The next day, mice were trained by exposing them to two identical objects (A) for 10 min and their activity was recorded using AnyMaze. After 1.5 h of training, one of the objects was replaced with a novel object (B), and mice were exposed to both old (A) and novel (B) objects to test their short-term recognition memory. Similarly, mice were exposed to objects (A) and (B) 24 h after training to test their long-term recognition memory. Time spent with objects was recorded when mice sniff or touch the objects, but did not lean against and/or climb on the object.

### Morris water maze

The Morris water maze test was conducted as described previously [17], where the mice were placed in a 1.5 m diameter pool with 25 °C water. Spatial cues were displayed around the pool and the platform was submerged 1 cm below the water's surface. The mice were acclimated to the room for 30 min prior to testing and then trained to find the platform in four trials (90 s) a day for 4 consecutive days with a 15 min intertrial period. Their activity was recorded with AnyMaze. If the mice did not find the platform within the 90 s, they were gently guided onto the platform. Short or long-term spatial memory was tested by removing the platform on day 5 or day 12, respectively. The latency to find the platform over 4 days of training, and the percent time spent in each quadrant of the maze was recorded for the probe tests.

### Contextual fear conditioning

The mice were acclimated to the room for 30 min prior to testing and then placed in a 20 cm × 10 cm enclosure with a metal grid floor connected to a shock generator. One wall of the enclosure was marked with a stripe pattern and mouse activity was recorded using AnyMaze. For training, mice were placed in the enclosure and allowed to freely explore it. After 150 s, a shock was administered (2 mA, 180 V, 2 s), and the mice were returned to their home cage after 30 s. After 24 h, contextual fear memory was tested by placing the mice in the enclosure for 6 min. Freezing time was reported in 30 s intervals, where freezing was defined as immobility lasting more than 0.5 s.

### Statistical analysis

The Student $T$ test (unpaired, two-tailed) or one-way ANOVA for experiments with one variable or two-way repeated-measures ANOVA with post hoc test for experiments with two variables were performed using GraphPad Prism software (GraphPad Software Inc, California, USA). All results are depicted as mean ± SEM unless indicated otherwise. $P$-values of less than 0.05 were considered to indicate significance. All statistical details are outlined in the figure legends, except for behavior tests, where statistics are provided in S12 Table.

### Supporting information

**S1 Fig. Morphology, density, and CD68 immunostaining in control and Atrx miKO microglia. (A)** Weight of 2- and 3-month-old control and ATRX miKO mice (2 months, CTL $n = 11$, ATRX miKO $n = 7$, $p = 0.080$; 3 months, CTL $n = 6$, ATRX miKO $n = 6$, $p = 0.679$ Student $T$ test). **(B)** RT-qPCR of Cre and Sun1GFP transcripts in 2-month-old control and ATRX miKO mice ($n = 4$ each genotype, Sun1GFP $p = 0.871$; Cre $p = 0.804$, Student $T$ test). Results were normalized to beta-actin transcript levels. **(C)** Quantification of immunofluorescence staining of Sun1GFP and IBA1 reveals >95% Cre expression in control and ATRX miKO mice across different brain regions ($n = 3$ each genotype, cortex $p = 0.423$, hippocampus $p = 0.351$, cerebellum $p = 0.119$, striatum $p = 0.327$). **(D)** Representative images of Ai14$^+$ cells with round, medium, and elongated soma. **(E, F)** Quantification of soma area and aspect ratio in the cortex at 2- and 3-months of age ($n = 3$ each genotype, soma area 2 months $p = 0.0003$, 3 months $p = 0.016$; aspect ratio 2 months $p = 0.029$, 3 months $p = 0.007$, Student $T$ test). **(G, H)** Quantification of nuclear area and aspect ratio in the cortex at 2- and 3-months ($n = 3$ each

genotype, nuclear area 2 months $p = 0.001$, 3 months $p = 0.034$; aspect ratio 2 months $p = 0.0937$, 3 months $p = 0.005$, Student $T$ test). **(I)** Quantification of elongated soma of microglia in hippocampal CA2, CA3, and DG of 2- and 3-months-old mice ($n = 3$ each genotype, CA2 elongated $p = 0.027$; CA3 elongated $p = 0.002$; DG elongated $p = 0.018$, Student $T$ test). **(J)** Quantification of CD68 foci per Ai14-labeled microglia in hippocampal CA2, CA3, and DG of 2-, 3-, and 6-month-old mice ($n = 3$ each genotype, 2 months CA2 $p = 0.861$, 3 months CA2 $p = 0.006$, 6 months CA2 $p = 0.193$; 2 months CA3 $p = 0.931$, 3 months CA3 $p = 0.023$; 6 months CA3 $p = 0.067$; 2 months DG $p = 0.243$, 3 months DG $p = 0.002$, 6 months DG $p = 0.935$, Student $T$ test). **(K)** Microglia density in hippocampal CA2/3 and DG of 2-, 3-, and 6-month-old control and ATRX miKO mice ($n = 3$ each genotype, CA2/3 2 months $p = 0.975$, 3 months $p = 0.018$, 6 months $p = 0.451$; DG 2 months $p = 0.831$, 3 months $p = 0.007$, 6 months $p = 0.415$, Student $T$ test). In all panels, control data is shown in black and ATRX miKO data is shown in red. The data underlying this figure can be found in the S1 Data file.
(TIF)

**S2 Fig. Chromatin accessibility in microglia lacking ATRX.** **(A)** Genomic distribution of DARs called by csaw and MACS. **(B)** Overlap of DARs called by csaw and MACS. **(C)** Heatmaps representing an association between gene expression and chromatin accessibility for cell cycle and DNA repair pathway genes. The z-score was computed from the RNA expression or ATAC-seq signals. **(D)** Macro view of UCSC tracks for example LINE, LTR, and SINE from Fig 4G.
(TIF)

**S3 Fig. Comparison of TE subfamily expression in different cell types lacking ATRX.** Scatterplots of transposable element family expression differences between control and ATRX KO in **(A)** microglia, **(B)** oligodendrocyte precursor cells (OPCs), and **(C)** astrocytes. Significantly upregulated TE subfamiles are indicated in red and downregulated TE subfamiles are indicated in blue. TE subfamilies not displaying significant change between control and KO cells are shown in gray.
(TIF)

**S4 Fig. RIG1 and cGAS levels in different brain regions. (A)** Expression of genes in the RIG1 and cGAS pathways. Variance stabilizing transformation (VST) counts from RNA-seq data are shown. $n = 3$ each genotype. **(B)** Western blot analysis of RIG1 and cGAS in control and Atrx miKO cortex and hippocampus at 3 months. Quantification is shown in graphs on the right ($n = 3$ each genotype). **(C)** Western blot analysis of RIG1 and cGAS in control and Atrx miKO cerebellum at 2 and 3 months. Error bars represent ±SEM. $n = 3$ each genotype. Unpaired Student $T$ test. The data underlying this figure can be found in the S1 Data file.
(TIF)

**S5 Fig. Effects of ATRX microglial deficiency on the dentate gyrus.** Quantification of **(A)** Ki67$^+$ proliferating cells and **(B)** DCX$^+$ differentiating cells in the dentate gyrus of control and ATRX miKO mice ($n = 5$ control and $n = 6$ KO mice). **(C)** Immunofluorescence staining of dentate gyrus for activated caspase 3 reveals no difference in apoptosis between control and ATRX miKO mice ($n = 3$ each genotype). The data underlying this figure can be found in the S1 Data file.
(TIF)

**S6 Fig. Supplemental data related to behavior tests. (A)** The number of entries in the open and closed arms or in the center of the elevated plus maze over 5 min (open arm entries $p = 0.710$, closed arm entries $p = 0.986$, center arm entries $p = 0.577$, Student $T$ test). Error bars represent ± SEM. **(B)** Distance traveled and **(C)** swimming speed over 4 days of training (4 trials/day) in the Morris water maze task (distance traveled $F(1, 33) = 0.5162$, $p = 0.477$; swimming speed $F(1, 132) = 0.1696$, $p = 0.681$, two-way ANOVA). **(D)** Discrimination index at 1.5 h ($t = 0.5848$, df = 22, $p = 0.5646$) and 24 h ($t = 1.696$, df = 21, $p = 0.1048$) probe test of the novel object recognition test. The data underlying this figure can be found in the S1 Data file.
(TIF)

**S1 Table. List of differentially expressed genes (DEGs).** RNA-sequencing data of Sun1GFP⁺ microglia nuclei obtained from the cortex and hippocampus of control and ATRX miKO mice by fluorescence-activated nuclei sorting (FANS).
(XLSX)

**S2 Table. Gene Set Enrichment Analysis (GSEA) categories.** Categories identified by GSEA of upregulated and down-regulated DEGs identified from RNA-sequencing analysis of control and ATRX miKO microglia nuclei.
(XLSX)

**S3 Table. List of differentially accessible regions (DARs).** DARs identified in microglia nuclei sorted from control and ATRX miKO cortex and hippocampus and subjected to the assay for transposase-accessible chromatin followed by sequencing (ATAC-seq).
(XLSX)

**S4 Table. List of top TFs with altered binding scores.** ATAC-seq data from sorted control and ATRX-null microglia nuclei was analyzed to identify the top represented transcription factors binding sites displaying differential binding scores (top 2%) using TOBIAS.
(XLSX)

**S5 Table. List of DARs identified in repetitive sequences.** DARs identified using the ATAC-seq data in repetitive sequences across the genome between control and ATRX-null microglia.
(XLSX)

**S6 Table. List of altered TE transcripts in ATRX-null microglia.** Differentially expressed retroelements identified in RNA-sequencing data of control and ATRX-null microglia. RetroTector analysis was applied to categorize these elements as full-length ERVs or solo LTRs.
(XLSX)

**S7 Table. TE subfamily analysis in microglia.** List of altered TE subfamilies in ATRX-null microglia.
(XLSX)

**S8 Table. TE subfamily analysis in astrocytes and oligodendrocytes.** List of altered TE subfamilies in ATRX-null oligodendrocyte precursor cells and astrocyte.
(XLS)

**S9 Table. List of bidirectionally expressed TEs.** Significantly differentially expressed TEs were identified for forward and reverse analysis and TEs expressed from both strands were identified.
(XLS)

**S10 Table. Cytokine/chemokine array and expression data.** Results of cytokine/chemokine array performed using cortical protein extracts from control and ATRX miKO mice as well as the RNA-seq data.
(XLSX)

**S11 Table. Oligonucleotides.** List of oligonucleotides used in this study.
(XLSX)

**S12 Table. Statistics.** Details of statistical analyses applied for different behavior tests.
(DOCX)

**S13 Table. Mapping details.** Additional information on the number of assigned, unmapped, and duplicate sequencing reads.
(XLS)

**S1 Data. Source data file.** File containing raw data for experiments depicted in the main figures and supplementary figures.
(XLSX)

**S1 Raw Images. Supplementary images of western blots and agarose gels.** File containing raw images for the corresponding cropped images in Figs 4 and S4.
(PDF)

## Acknowledgments

We are grateful to Vania Prado and Marco Prado for the Cx3Cr1[ER] mice and Samuel Asfaha for access to the Sony SH800 sorter. The behavioral assays were performed at the Robarts Research Institute neurobehavioral core facility and next-generation sequencing at Canada's Michael Smith Genome Sciences Centre (BC Cancer Research, Vancouver, BC, Canada). Fig 1A was partially created in BioRender. Berube, N. (2025) https://BioRender.com/o16yx27. Fig 2A was created in BioRender. Shafiq, S. (2025) https://BioRender.com/svcdvbl.

## Author contributions

**Conceptualization:** Sarfraz Shafiq, Wataru Inoue, Nathalie G. Bérubé.

**Data curation:** Milica Pavlovic.

**Formal analysis:** Sarfraz Shafiq, Alireza Ghahramani, Kasha Mansour, Milica Pavlovic.

**Funding acquisition:** Nathalie G. Bérubé.

**Investigation:** Sarfraz Shafiq, Kasha Mansour, Miguel Pena-Ortiz, Julia K. Sunstrum, Yan Jiang, Megan E. Rowland, Nathalie G. Bérubé.

**Methodology:** Sarfraz Shafiq, Alireza Ghahramani, Miguel Pena-Ortiz, Julia K Sunstrum, Milica Pavlovic, Wataru Inoue, Nathalie G. Bérubé.

**Project administration:** Nathalie G. Bérubé.

**Resources:** Wataru Inoue, Nathalie G. Bérubé.

**Supervision:** Nathalie G. Bérubé.

**Visualization:** Alireza Ghahramani, Kasha Mansour, Julia K. Sunstrum, Milica Pavlovic, Nathalie G. Bérubé.

**Writing – original draft:** Sarfraz Shafiq, Nathalie G. Bérubé.

**Writing – review & editing:** Sarfraz Shafiq, Alireza Ghahramani, Kasha Mansour, Miguel Pena-Ortiz, Julia K. Sunstrum, Yan Jiang, Megan E. Rowland, Wataru Inoue, Nathalie G. Bérubé.

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
