## [Editor Report · Decision Letter 0]

6 May 2024

Dear Dr Bérubé,

Thank you for submitting your manuscript entitled "Viral mimicry and memory deficits upon microglial deletion of ATRX" for consideration as a Research Article by PLOS Biology.

Your manuscript has now been evaluated by the PLOS Biology editorial staff as well as by an academic editor with relevant expertise and I am writing to let you know that we would like to send your submission out for external peer review.

Once your full submission is complete, your paper will undergo a series of checks in preparation for peer review. After your manuscript has passed the checks it will be sent out for review. To provide the metadata for your submission, please Login to Editorial Manager (https://www.editorialmanager.com/pbiology) within two working days, i.e. by May 08 2024 11:59PM.

Kind regards,

Luke

Lucas Smith, Ph.D.

Senior Editor

PLOS Biology

lsmith@plos.org

---

## [Decision Letter · Decision Letter 1]

23 Jul 2024

Dear Dr Bérubé,

Thank you for your patience while your manuscript "Viral mimicry and memory deficits upon microglial deletion of ATRX" was peer-reviewed at PLOS Biology. Your manuscript has been evaluated by the PLOS Biology editors, an Academic Editor with relevant expertise, and by several independent reviewers.

As you will see, the reviewers thought the study was interesting, but had some concerns. Both R2 and R3 thought the microglia morphological analysis can be improved. More critically, R1 had serious concerns about the bioinformatic analysis of the TEs. Based on their specific comments and following discussion with the Academic Editor, it is clear that a substantial amount of work would be required to meet the criteria for publication in PLOS Biology. However, given our and the reviewer interest in your study, we would be open to inviting a comprehensive revision of the study that thoroughly addresses all the reviewers' comments. Given the extent of revision that would be needed, we cannot make a decision about publication until we have seen the revised manuscript and your response to the reviewers' comments. Your revised manuscript would need to be seen by the reviewers again, but please note that we would not engage them unless their main concerns have been addressed.

Having discussed the reviews with the Academic Editor, we think you should address all concerns. Specifically, please address the concerns about the morphological characterization of the microglia, and we will need to see an improved bioinformatic analysis of the TEs, and whether the effects are truly TE-mediated, in order to consider a revised version of this manuscript.

We appreciate that these requests represent a great deal of extra work, and we are willing to relax our standard revision time to allow you 6 months to revise your study. Please email us (plosbiology@plos.org) if you have any questions or concerns, or envision needing a (short) extension.

**IMPORTANT - SUBMITTING YOUR REVISION**

*Resubmission Checklist*

*Published Peer Review*

*PLOS Data Policy*

*Blot and Gel Data Policy*

Sincerely,

Suzanne

Suzanne De Bruijn, PhD,

Associate Editor

PLOS Biology

sbruijn@plos.org

REVIEWS:

Reviewer #1: Identified himself as Johan Jakobsson.

In the study "Viral mimicry and memory deficits upon microglial deletion of ATRX", Shafiq et al. characterize the loss of ATRX in mice microglia. ATRX is an important chromatin remodeler whose mutations have been implicated in human intellectual disability. They report that the loss of ATRX in mice microglia triggers viral mimicry, which is linked to cell-extrinsic effects and behavioural changes. They suggest this might be due to a transcriptional activation of TEs.

Many parts of this manuscript are interesting and well done. The observations about immune response and the cell-extrinsic effects on neurons are well-conducted and interesting. The behavioural changes—although sometimes inconsistent—are intriguing. However, I have serious concerns about the suggestion that this is TE-mediated.

The idea that TEs are upregulated in ATRX mice is interesting. However, I have serious concerns about the robustness of this result. Bioinformatical analysis of TE expression in mice is very challenging and requires the use of dedicated pipelines. I would like to recommend the authors to revisit this analysis, perhaps with help from someone who is an expert in the field.

For example, the authors do not make a distinction between proviruses and solo LTRs, which is crucial for the suggestion of an ERV-induced viral mimicry. This distinction would greatly improve the precision of the study's conclusions.

They highlight three LTR families, but the LINE and SINE classes are presented as whole classes (classes that are equally or even more diverse than LTRs). While some of their subfamilies are evolutionary young, the vast majority of the elements in these classes are degenerated copies—therefore, it is crucial to be more specific about which subfamilies are found to be upregulated and how intact these individual elements are. As it is now, all their observations could be based on noice.

Figure 4D looks strange - is a threshold missing from the title/legend? How is it possible that NO elements are left unchanged? The same question about the ATAC-seq results and Figure S2D. The authors should reassure the reader that this is not noise. One could show a more global analysis to prove this is not an artefact, for example, with housekeeping genes or describing the ATAC-seq peaks that do not show an effect.

In addition, the authors claim that the immune response is triggered by the formation of double stranded RNAs derived from TEs. I don't think this is shown by the Strand-specific analysis of the RNA-seq data shown in Figure 4G as they say. Additional experiments are needed to prove the presence of dsRNAs in microglia, as well as more specific characterization of the type of TEs from which these could be transcribed.

In addition, the authors could perform immunostainings for retroviral proteins (e.g. IAP-gag, if these subfamilies are indeed the ones upregulated in the ERVK family).

The SINE and LINE examples shown in Figure 4G look like noise or long non-coding RNAs. I encourage them to show a zoom-out with the transcription start site, include the subfamily they are referring to, length, and evolutionary age / conservation.

Finally, the authors should discuss the similarities, but also stark differences between mice and human TE composition.

In summary, this is an interesting manuscript - but the TE-analysis needs to be greatly improved.

Reviewer #2:

The authors have previously explored the role of ATRX in neurons on cognitive function and other relevant endpoints. The present manuscript attempts to extend this by deleting it specifically within microglia using inducible approach, finding that its loss in microglia alters microglial state, chromatin accessibility, and various transcriptomic profiles in a way that is associated with an immune response that may activate cGAS and related components to lead to upregulation of cytokines and ultimately cognitive dysfunction. The study is overall well-designed and presents several interesting results. However, there are a number of issues with interpretation and rigor that limit overall enthusiasm somewhat, as outlined below:

1. The authors should rephrase the framing of "activated microglia" throughout the manuscript. Our understanding of what makes microglia activated and how this contributes to disease has matured significantly and has become more nuanced. For example, many of these so-called "activation" markers are expressed in cells exhibiting protective phenotypes as well as cells that have become senescent in some states. Defining a binary of activated or not is likely not accurate.

2. There is a peculiar loss of CD68 per cell between 2 and 3 months of age. This hasn't been assessed statistically (but should be). This raises concern about the method being used given that CD68 generally increases with age. The authors should account for why there is this difference, if significant, as well as the decrease in microglial density of microglia (in hippocampus). Does this trend continue in later life? Moreover, the decrease microglial density in Fig S1 is represented as a violin plot whereas all other data are simple bar scatterplots. It's unclear whether this is an example of pseudoreplication where all cells in the groups were compared to one another or whether averages of cell counts in sections for each mouse were compared as appropriate. This should be clarified.

3. The authors should examine which genes among their DEGs presented for highlighted pathways are those that have been previously characterized in the literature to change in response to microglial isolation procedures like FANS. Deletion of ATRX could intersect with these artifactual changes in response to isolation and affect the pathways highlighted.

4. The choice of 2 vs 3 months for ages of study should be justified. No rationale for this is given, and it's unclear what phenotypes would be expected at developmental timepoints and later age timepoints based on the results shown. As the introduction highlights the importance of this gene for intellectual disability, it seems necessary to justify ages and why deletion wouldn't make sense at an earlier timepoint prior to adulthood (2 and 3 months).

5. The statement that multiple genes belonging to DNA/RNA sensing pathways are overexpressed in ATRX-null microglia from RNAs-seq (lines 190-191) should be substantiated with enrichment analyses and graphs of the individual genes (for supplement). This is key to the argument being made so "data not shown" is not sufficient evidence.

6. How generalizable is the elevation of cGAS? Is this also true in hippocampus? Is it consistent across ages? Also, immunoblotting experiments showing elevations of DNA/RNA sensing components (cGAS, RIG-1, STAT1, etc) are all performed in whole cortex tissue, yet the change is presumably only in microglia, one portion of the total pool in a lysate. Does dysfunction in microglia lead to similar changes in neurons, which is driving the effects, or is it specific to the microglia? How do isolated microglia look for these components?

7. Several interpretations are often heavy-handed. For example, the authors link cytokine elevations to the antiviral immune response and cGAS the identified, but many pathways they reveal from pathways analysis of DEGs imply that many causes may lead to these changes. Moreover, the authors have not shown whether this deletion causes astrocytosis as a consequence, which could also lead to increases in cytokines.

8. The issue in point #7 occurs also in the discussion when the authors argue that the cognitive dysfunction likely stems from the cytokine elevations when they have shown data that neuronal excitability is somehow affected (which likely also affects cognition). Moreover, the discussion fails to provide a satisfying explanation for how microglial deletion of ATRX affects neuronal excitability.

9. What is the basis on which each of the subclusters is annotated? Can deconvolution be a robust method here to create subclusters if the KO creates significant shifts in microglial state that leads to differing populations not present in the original single cell dataset?

10. The authors make a statement that all data underlying the manuscript are available in the supporting documents and reporting checklist and not upon request. Therefore, it is confusing that there are multiple instances of "data not shown" throughout the manuscript. As one example, one of the key issues is whether other cell types are affected by microglial deletion of ATRX (and how), yet this is not addressed aside from a statement that they looked at astrocytes and oligodendrocytes, which is not shown.

11. In Figure 6, J through K should also be shown along with discrimination index. Importantly, there appears to be no multiple comparison testing with post tests. Only t tests appear to be used here, which is not correct. This should be corrected throughout the manuscript where it occurs.

Reviewer #3:

This manuscript by Shaqif et al., entitled "Viral mimicracy and memory deficits upon microglial deletion of ATRX" , shows that the conditional ablation of chromatin remodeler ATRX gene (mutations in this gene are known to cause alpha-thalassemia and mental retardation) in microglia leads to de-repression of endogenous retroelements (e.g. ERVs) triggering virus mimicry, which in turn activates RNA and DNA sensing pathways (e.g. RIG-1 which senses short dsRNA) that presumably converge at the mitochondrial interface to activate MAVS, and TBK-1 induced I3F1 phosphorylation resulting in interferon-mediated neuroinflammation and cognitive deficits. The authors posit that such deficits can potentially contribute to the pathology caused by ATRX in the nervous system. In line with this they propose that potential therapeutic approaches should consider repressing retroelements and nucleic acid sensing, or suppressing microgial immune activation.

I find that this manuscript reads well, the findings are presented with clarity, the methods used are broad (including RNAseq of FACS-isolated nuclei, ATAQ-seq, electrophysiology and behaviour testing) and adequate, and the experimental data mostly support the conclusions made by the authors. Their data also further reinforces the emergent view that microglia are key regulators of synaptic function and that microglia dysfunction might play a central roles in the onset and/or progression of different neurological disorders.

I have only a few minor points:

* The categorization of microglia shape suggests changes in microglia reactivity triggered by the loss of ATRX. Together with the CD68 IHC and transcriptomic data the authors conclude that microglia are reactive. It maybe so, however the quality of the microglia morphological characterization does not match the general quality of the paper. The authors should consider in illustrating such changes in microglia morphology with good quality and higher magnification representative pictures (for example in Fig1) . In addition, Scholl analyses of microglia would provide a much better (and quantitative) assessment of changes in microglia morphology.

* In Figure1 E in the Y axis - correct "elogated" to elongated

* In Figure 2 F is difficult (at least in my PDF) to really identify Ki67 + cells. Also the clear difference in proliferation shown in the graphic between control and ATRX miKO at 2 months is not evident in the figure. An inset with higher magnification would help.

---

## [Decision Letter · Decision Letter 2]

30 May 2025

Dear Dr Bérubé,

Thank you for your patience while we considered your revised manuscript "Viral mimicry and memory deficits upon microglial deletion of ATRX" for consideration as a Research Article at PLOS Biology. Your revised study has now been evaluated by the PLOS Biology editors, the Academic Editor and by two of the original reviewers.

As you will see in their comments, below, while both reviewers think that the paper has been strengthened in the revision, they have identified a number of lingering issues which we think should be addressed in another revision. While we think that many of the last points will be relatively straightforward to address, we do think that it will be important to provide the additional analyses requested by Reviewer 1. We also would like to emphasize the requests to provide the underlying code to github (and with a zenodo DOI), as is PLOS policy, and to incorporate relevant responses from the response to reviewers into the manuscript as suggested by Reviewer 2. As a editorial note, reviewer 2 has raised concerns with the statistics in the behavioral experiments, and we largely agree with his/her concerns. We think that statistical comparisons between groups will be necessary to make claims about the effects of ATRX on memory. Without stronger statistical support for those differences, we think that the claims about memory and cognition will need to be toned down throughout the manuscript, and in the title. After discussion within the team, we think the paper would still be suitable for publication if the claims about cognition are tempered, as we think the finding that loss of ATRX induces a viral mimicry response is still interesting.

We expect to receive your revised manuscript within 1 month, but please email us (plosbiology@plos.org) if you have any questions or concerns, or would like to request an extension.

We will then assess your revised manuscript and your response to the reviewers' comments with our Academic Editor aiming to avoid further rounds of peer-review, although we might need to consult with the reviewers, depending on the nature of the revisions.

**IMPORTANT - SUBMITTING YOUR REVISION**

*Resubmission Checklist*

*Published Peer Review*

*PLOS Data Policy*

*Blot and Gel Data Policy*

Sincerely,

Luke

Lucas Smith, Ph.D.

Senior Editor

PLOS Biology

lsmith@plos.org

REVIEWS:

Reviewer #1, Johan Jakobsson (note, reviewer 1 has signed this review): The revised manuscript by Shafiq et al. has been greatly improved and is now almost ready for publication. This study will be of great interest to the field as it opens up new ideas regarding the link between neurodevelopmental disorders, neuroinflammation and transposable element activation. We only have a few minor comments that we suggest the authors address to clarify some details in the manuscript before publication.

1. An internal region does not necessarily mean that the ERV is a provirus or that it is "full-length". There are tools to better classify ERVs' repeatmasker annotation (such as Retrotector, as per Jönsson et al, 2021) to properly stitch together the commonly fragmented internal regions, and separately annotated LTRs, to their corresponding internal regions. The authors should update their analysis along these lines regarding the expression of full-length ERVs since this is key for the interpretation of their data.

2. The authors should provide the relevant scripts in a git repository in order for the readers and reviewers to evaluate the quality of the analysis. The code should also be well documented. Please provide the relevant GitHub repository, preferably with a zenodo DOI.

3. We appreciate the discussion section highlighting the differences between the human and mouse TE landscape. However, the section is incomplete and should be expanded. For example, a key piece of information that is missing is that mice have ERVs that can retrotranspose while humans do not. This is likely to have a major impact on how these elements are controlled and the consequences of their activation. This should be discussed.

4. Figure S1F y-axis label is overlapping the numbers.

5. Figure 4B,D should show the number of elements being up/down regulated (and preferably the total. E.g. 20/1000)

6. Figure 4H has an illustrator text box showing "Lorem ipsum".

Reviewer #2: The authors have addressed several of the previous concerns. However, there are remaining concerns to address that affect interpretations of the study.

1. in many cases, legitimate concerns have been raised and then addressed thoroughly in the rebuttal text, but these responses have not been described in the discussion or in supplementary data where they could benefit the manuscript in a substantial way. See examples for the second part of point #6, etc and other examples throughout responses to previous concerns.

2. In Figure 6, several behavioral analyses (6I and 6k) are showing differences in performance only within a genotype but inferring differences across genotypes based on lack of difference within one of the genotypes. This type of "implicit" comparison as an approach is a statistical error and lacks rigor, as described in detail previously (https://www.nature.com/articles/nn.2886). Also, why is a Violin plot provided for 6C and D? Are these showing individual mouse replicates or comparisons across multiple trials? The former should be shown as a scatterplot.

3. Somewhat related to the previous point, but the behavioral results overall are somewhat problematic in how they're being interpreted - there is not solid evidence that there are cognitive effects of sufficient quality to argue functional changes here. After requesting the discrimination index to be reported in Figure 6, this has now been included, though it is not commented upon that the effect is not significant, supporting the overall trend that there are not robust cognitive changes following manipulation of ATRX. The question raised now is whether the mechanism put forth by the authors requires cognitive changes for it to be valid. Overall, the behavioral data argue that the effect of ATRX modulation is not strongly linked to cognitive changes. This has not been placed in the context of the model being proposed in the paper.

---

## [Decision Letter · Decision Letter 3]

29 Jul 2025

Dear Dr Bérubé,

Thank you for your patience while we considered your revised manuscript "Microglial ATRX deficiency elicits a viral mimicry response and impairs hippocampal-dependent memory" for publication as a Research Article at PLOS Biology. This revised version of your manuscript has been evaluated by the PLOS Biology editors, the Academic Editor and by two of the original reviewers.

As you will see in the reviewer comments, which are appended below, both reviewers 1 and 2 think the revision has largely addressed the previous issues. However the reviewers also have a few lingering concerns that we think should be addressed, in a last revision that we anticipate will not take very long. I note that reviewer 2 has commented that, after applying the new statistics, the behavioral data is not sufficiently robust to make claims about hippocampal dependent memory in the title - and we agree with this reviewer. We are OK with the paper interpreting the morris water maze data as a change in memory in the main text, as this is appropriately discussed with the relevant nuance - but we think that claims about memory should be removed from the title and from the abstract.

Along those lines, we would propose that the title of your paper be changed to "Microglial deficiency in the ATRX chromatin remodeler elicits a viral mimicry immune response that impacts neuronal function and behavior", as I think would captures the subtle effect in memory, and also the anxiety phenotypes reported here.

Based on the reviews, we are likely to accept this manuscript for publication, provided you satisfactorily address the remaining points raised by the reviewers. Please also make sure to address the following data and other policy-related requests.

**IMPORTANT: Please address the following editorial requests.

1) FINANCIAL DISCLOSURES: Please update the financial disclosures statement in our editorial manager system, to include the grant numbers for all grants and the URL of each funder website.

2) ETHICS STATEMENT: Please update your ethics statement, in your methods section, to include the approval number for your animal protocol approved by the University of Western Ontario IACUC.

3) DATA: Thank you for providing the ATAC and RNA seq data on SRA. For some reason, I could not access this data with the accession number provided. Can you please ensure the accession number included in the data availability statement is correct (PRJNA787973) and that the data is publicly available?

4) DATA: Thanks also for providing a supplemental excel file with the underlying data for your study. Please add a sentence to each figure legend, directing readers to the underlying data. Please also add a sentence to the data availability statement, in our editorial manager system pointing to this file.

5) CODE: Thank you also for depositing the scripts used in this study on GitHub and including a DOI from Zenodo. Please add this information in the data availability statement in our editorial manager system - as that is the version that will be published with the paper.

6) BLOT AND GEL REPORTING REQUIREMENTS: Please note that we require the original, uncropped and minimally adjusted images supporting all blot and gel results reported in an article's figures or Supporting Information files. We will require these files before a manuscript can be accepted so please prepare and upload them now. Please carefully read our guidelines for how to prepare and upload this data: https://journals.plos.org/plosbiology/s/figures#loc-blot-and-gel-reporting-requirements

7) DATA NOT SHOWN: Please note that per journal policy, we do not allow the mention of "data not shown", "personal communication", "manuscript in preparation" or other references to data that is not publicly available or contained within this manuscript. Please either remove mention of these data or provide figures presenting the results and the data underlying the figure(s).

We expect to receive your revised manuscript within two weeks.

*Published Peer Review History*

*Press*

Sincerely,

Luke

Lucas Smith, Ph.D.

Senior Editor

lsmith@plos.org

PLOS Biology

Reviewer remarks:

Reviewer #1, Johan Jakobsson and Raquel Garza (note, Drs Jakobsson and Garza reviewed this study together, and provided comments as reviewer 1. They have opted to sign this review):

Overall, the comments regarding ERV classification and interpretation of the results are much better and the manuscript is now ready for publication. However, a couple of details that could help the reader are still missing:

1. The methods section described the stringency of the mapping of the reads for the multi-mapping for TEtranscripts quantification, but the authors haven't specified the parameters used for the mapping that was used for the per-element differential expression analysis or, e.g. the tracks in Figure 4H.

2. This is equally or more important for the ATACseq. In general, it would be good for the authors to give some details on the mappability of the reads to these elements for the both of the sequencing technologies.

3. "LTR-type" should be rephrased to "LTR class" (same with LINE and SINE type —> class). Assuming that the figure is indeed showing LTR families.

4. "In contrast, most human ERVs are highly mutated and lack retrotranspositional capacity" No HERV has been proven to retrotranspose in the human genome.

Reviewer #2:

Overall, the authors have sufficiently addressed concerns for all rounds of review. The only lingering issue relates to the interpretation of hippocampal-dependent memory in the manuscript. While the authors acknowledge that there are no true changes in NOR data in the mice that reached statistically significance to warrant claims that the mice have functional differences, the authors show a significant difference in MWM data on the 12th day of testing that does reach significance. Overall, the spatial memory results in the manuscript are not sufficiently robust to make the claim there is a change in hippocampal spatial memory. Multiple assays would be needed to make this claim, and showing a difference at one timepoint in one assay does not support this claim overall. This was adequately addressed in the text, but the limitation should be addressed in the title by removing the reference to memory.

---

## [Editor Report · Decision Letter 4]

15 Aug 2025

Dear Nathalie,

Thank you for the submission of your revised Research Article "Microglial deficiency in the ATRX chromatin remodeler elicits a viral mimicry immune response that impacts neuronal function and behavior" for publication in PLOS Biology and thank you for addressing the last editorial and reviewer requests in this revision. On behalf of my colleagues and the Academic Editor, Mikael Simons, I am pleased to say that we can in principle accept your manuscript for publication, provided you address any remaining formatting and reporting issues. These will be detailed in an email you should receive within 2-3 business days from our colleagues in the journal operations team; no action is required from you until then. Please note that we will not be able to formally accept your manuscript and schedule it for publication until you have completed any requested changes.

**Please note - as discussed over email, I have updated your manuscript file and raw images file with the new versions that you provided me. Please do take a moment to double check that everything looks OK after these changes (and as an FYI - I turned the sentence that you highlighted in blue in your manuscript back to black).

PRESS

We frequently collaborate with press offices. If your institution or institutions have a press office, please notify them about your upcoming paper at this point, to enable them to help maximize its impact. If the press office is planning to promote your findings, we would be grateful if they could coordinate with biologypress@plos.org. If you have previously opted in to the early version process, we ask that you notify us immediately of any press plans so that we may opt out on your behalf.

Sincerely, 

Luke

Lucas Smith, Ph.D.

Senior Editor

PLOS Biology

lsmith@plos.org